# Improving Energy Natural Gradient Descent through Woodbury, Momentum, and Randomization

**Andrés Guzmán-Cordero**
Vector Institute, Mila - Quebec AI Institute, Université de Montréal
andres.guzman-cordero@mila.quebec

**Felix Dangel**
Vector Institute
f.dangel@vectorinstitute.ai

**Gil Goldshlager**
UC Berkeley
ggoldsh@berkeley.edu

**Marius Zeinhofer**
ETH Zurich
marius.zeinhofer@sam.math.ethz.ch

## Abstract

Natural gradient methods significantly accelerate the training of Physics-Informed Neural Networks (PINNs), but are often prohibitively costly. We introduce a suite of techniques to improve the accuracy and efficiency of energy natural gradient descent (ENGD) for PINNs. First, we leverage the Woodbury formula to dramatically reduce the computational complexity of ENGD. Second, we adapt the Subsampled Projected-Increment Natural Gradient Descent algorithm from the variational Monte Carlo literature to accelerate the convergence. Third, we explore the use of randomized algorithms to further reduce the computational cost in the case of large batch sizes. We find that randomization accelerates progress in the early stages of training for low-dimensional problems, and we identify key barriers to attaining acceleration in other scenarios. Our numerical experiments demonstrate that our methods outperform previous approaches, achieving the same $L^2$ error as the original ENGD up to $75\times$ faster.

## 1 Introduction

Using neural networks to solve partial differential equations (PDEs) is a promising and highly active research direction at the intersection of machine learning and scientific computing. While the idea is not new [7, 20], it has gained traction with the advent of Physics Informed Neural Networks (PINNs) [33]. Compared to traditional methods, neural networks avoid the need for sophisticated, problem-dependent discretization schemes and promise to scale better to high-dimensional problems [33]. However, it is very challenging to attain the accuracy levels of traditional solvers with neural networks. This is in large part due to the loss landscape's non-convexity and ill-conditioning [44]. To address this problem, second-order methods such as energy natural gradient descent [ENGD, 27] and Kronecker-factored approximate curvature [KFAC, 24, 6] have been proposed. These methods greatly improve upon the accuracies attainable by first-order optimizers like SGD or Adam [6], but they suffer from either a high per-iteration cost or a high implementation complexity. As a result, there remains a significant need for more accurate and efficient methods to train PINNs.

In parallel to the PINN community, neural networks have also been leveraged in conjunction with the variational Monte Carlo method [VMC, 9] to simulate quantum many-body systems [3, 14, 21, 32].

39th Conference on Neural Information Processing Systems (NeurIPS 2025).

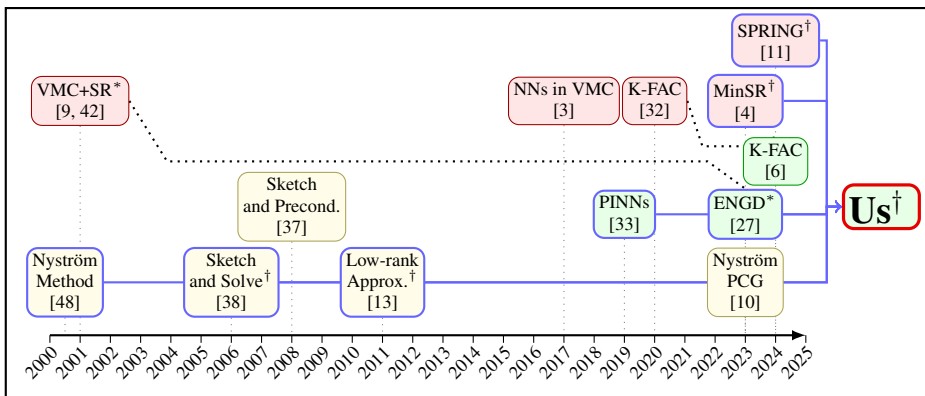

Figure 1: **Timeline of VMC, PINN and RLA methods**. $*$: Stochastic reconfiguration (SR) and energy natural gradient descent (ENGD) both precondition their stochastic gradients with the inverse of an appropriate curvature matrix. $\dagger$: Inspired by MinSR, we apply Woodbury's matrix identity to use the kernel matrix instead of ENGD's Gauss-Newton matrix, thus reducing the cost to $O(N^2 P)$ instead of $O(P^3)$, where $N$ denotes the batch size and $P$ the number of parameters; we further introduce SPRING for PINNs to transport curvature information across optimization steps; last, we use a GPU-efficient Nyström approximation to reduce the iteration cost for large batch sizes.

Second-order optimizers are very common within this field and the most popular one, known as stochastic reconfiguration [SR, 42, 1], shares a similar computational structure to ENGD, owing to a similar mathematical derivation as a projected functional algorithm [28]. Within the neural network VMC community, the quest to model complex quantum systems to as high of an accuracy as possible has led to significant advances in algorithmic implementations of the SR optimizer, including the use of the Woodbury matrix identity to reduce the computational cost of each optimization step [MinSR, 4] and the development of momentum schemes that can accelerate the convergence of natural gradient methods [SPRING, 11].

Given the similarities between VMC and PINNs it is natural to wonder if PINNs can benefit from the techniques developed in the more mature optimization literature for VMC. In this paper, we answer this question in the affirmative (See Figure 1). We first show that both MinSR and SPRING can be adapted to drastically improve ENGD for PINNs. Going beyond transfer, we explore the use of randomized methods to further reduce the cost of both ENGD and SPRING iterations. Concretely, our contributions are as follows:

- **We transfer ideas from the VMC community to PINNs**, showing that the Woodbury matrix identity drastically improves the computational bottleneck of ENGD for PINNs. Concretely (details in Section 3), we compute in sample space, as its dimensionality is typically *much smaller* than the parameter space. This enables an exact ENGD for PINNs with up to $10^6$ trainable weights on standard consumer hardware with drastically improved computation time. Unlike K-FAC, the implementation is architecture-agnostic and thus highly flexible. We then show that the SPRING algorithm introduced in the VMC community can further improve the training process of PINNs, yielding consistently faster convergence and removing the need for the expensive line search from ENGD. The resulting method, summarized in algorithm 1, is straightforward to implement and achieves state-of-the-art results, and we thus recommend it as the new optimizer of choice for training PINNs.

- **Going beyond transfer, we explore the use of randomized algorithms** based on a variant of Nyström's method [10] to further accelerate both ENGD and SPRING in the case of large sample sizes. As standard formulations of Nyström's method require SVDs, which are extremely slow on GPU in contemporary ML libraries, we develop a GPU-efficient variant of Nyström that only uses Cholesky solves on square matrices of sketch dimension. We highlight this GPU-efficient Nyström approximation as an independent contribution which can be useful in a wide array of machine learning applications. For PINNs, we find that it can accelerate the early phases of training for low-dimensional problems, and we identify key barriers to extending this acceleration to the later phases and higher-dimensional settings.

- **We present thorough empirical evidence for our claims**, demonstrating the utility of the Woodbury identity and the SPRING algorithm as well as the power and the limitations of randomization. We explore both low- and high-dimensional problems and we meticulously tune the hyperparameters of each method to attain optimal performance, with final results supported by more than $4500$ training runs in total.

**Related work** The Woodbury identity has also been used to train neural networks for traditional machine learning problems [5, 49, 35] and other applications, such as model sparsification [41], involving the Fisher information matrix in traditional deep learning. Descriptions of the ill-conditioned landscape of PINNs and some solutions have been explored Rathore et al. [34]. While the key contribution of Rathore et al. [34] addresses the scalability of Newton's method to large neural networks and is thus similar to our results, their focus is on Newton's method rather than natural gradient methods. Kiyani et al. [18] evaluate quasi-Newton methods which leverage historical gradients for improved efficiency and accuracy across stiff and non-linear PDEs. Wang et al. [46] provides a theoretical framework for gradient alignment in multiobjective PINN training, demonstrating the potential of second-order methods to resolve directional conflicts through Hessian preconditioning.

## 2 Background

**Physics-Informed Neural Networks** PINNs reformulate forward or inverse PDE problems as least-squares minimization problems. Constraints, such as boundary conditions or data terms are typically included into the loss function as soft penalties. We illustrate the approach using a general PDE operator $\mathcal{L}$ and Dirichlet boundary conditions. Suppose we aim to solve the PDE

$$\mathcal{L}u = f \quad \text{in } \Omega,$$
$$u = g \quad \text{on } \partial\Omega.$$

Here $\Omega \subset \mathbb{R}^d$ is the computational domain (possibly including time), $f$ is a forcing term and $g$ are the boundary values. Introducing a neural network ansatz $u_\theta$ with trainable parameters $\theta \in \mathbb{R}^P$, the above equation is reformulated as a least-squares minimization problem

$$L(\theta) = \frac{|\Omega|}{2N_\Omega} \sum_{i=1}^{N_\Omega} (\mathcal{L}u_\theta(x_i) - f(x_i))^2 + \frac{|\partial\Omega|}{2N_{\partial\Omega}} \sum_{i=1}^{N_{\partial\Omega}} (u_\theta(x_i^b) - g(x_i^b))^2,$$

where $x_1, \ldots, x_{N_\Omega}$ are randomly drawn points in $\Omega$, and $x_1^b, \ldots, x_{N_{\partial\Omega}}^b$ are randomly drawn points on $\partial\Omega$. Note that if it holds $L(\theta) = 0$, then the PDE is satisfied at the points in the interior and the boundary conditions are met on the boundary points.

**Natural Gradient Methods for PINNs** It is well-known that PINNs are notoriously difficult to train with first-order methods [44, 19, 45]. Natural gradient methods present a promising alternative, able to achieve excellent accuracy far beyond what is possible with first-order methods or quasi-Newton methods [28, 27, 16, 6]. The natural gradient descent scheme introduced in [27], which we refer to as *energy natural gradient descent* is of the form

$$\theta_{k+1} = \theta_k - \eta_k (G(\theta_k) + \lambda I)^{-1} \nabla L(\theta_k), \quad k = 0, 1, 2, \ldots \tag{1}$$

where $\eta_k$ is a well chosen step-size and $\lambda > 0$ is a regularization parameter. The matrix $G(\theta_k)$ is the Riemannian metric induced by the functional Hessian in the tangent space of the ansatz, given by

$$G(\theta) = \frac{|\Omega|}{N_\Omega} \sum_{i=1}^{N_\Omega} \mathrm{J}_\theta \, \mathcal{L}u_\theta(x_i)^\top \, \mathrm{J}_\theta \, \mathcal{L}u_\theta(x_i) + \frac{|\partial\Omega|}{N_{\partial\Omega}} \sum_{n=1}^{N_{\partial\Omega}} \mathrm{J}_\theta \, u_\theta(x_n^b)^\top \, \mathrm{J}_\theta \, u_\theta(x_n^b), \tag{2}$$

where $\mathrm{J}_\theta$ denotes the Jacobian with respect to the trainable parameters. Here, we assumed the PDE operator $\mathcal{L}$ to be linear, for a nonlinear PDE operator we use its linearization, compare also to [16].

**Scalability of Natural Gradient Methods for PINNs** The matrix $G(\theta_k) \in \mathbb{R}^{P \times P}$ is quadratic in the number of trainable network parameters. Hence, naive approaches to solve $G(\theta_k)x = \nabla L(\theta_k)$ are doomed to fail for larger networks due to the cubic cost $\mathcal{O}(P^3)$. Strategies to treat the system in regimes where direct solvers are infeasible include matrix-free approaches based on the conjugate gradient method [16, 23] with matrix-vector products [31, 40], and approximating $G(\theta_k)$ with

an easy-to-invert Kronecker-factorization [6]. Both approaches have drawbacks. The matrix-free strategy suffers from the ill-conditioning of the matrix $G(\theta_k)$ [44] and can require large amounts of CG iterations to produce an acceptable solution to the linear system. The K-FAC approach [6] is complex and depends on the network architecture and the PDE. As described in detail in Section 3, we exploit a different structure in $G$ using a low-rank factorization present in the matrix $G(\theta_k)$, and we show that it yields a fast and accurate method to solve $G(\theta_k)x = \nabla L(\theta_k)$ that can naturally be combined with powerful techniques of randomized linear algebra.

**Variational Monte Carlo**   To improve the strategies of matrix-free solvers and K-FAC, we draw inspiration from the field of neural network wavefunctions, which leverages the variational Monte Carlo method [9] to simulate quantum systems from lattice models [3] to molecules and materials [14, 21]. Almost all works on neural network wavefunctions are based on an optimization approach known as stochastic reconfiguration (SR) [42], which is a quantum generalization of natural gradient descent (see [32], Appendix C). In fact, the same eq. (1) applies for SR, but in this case $G(\theta_k)$ is the Fisher information matrix. A number of works have demonstrated that such approaches outperform simpler first-order optimization strategies when training neural network wavefunctions [32, 22, 39]. Recently, Chen & Heyl [4] proposed the MinSR method to tackle the scalability of SR in the VMC context. The MinSR algorithm is able to directly calculate the SR search direction with a drastically reduced computational cost, relying on the fact that the rank of $G(\theta_k)$ is upper bounded by the batch size $N$. Even more recently, Goldshlager et al. [11] proposed the SPRING algorithm to accelerate the convergence of MinSR by incorporating a special form of momentum. In Section 3, we show how these methods can be applied in the context of PINNs.

**VMC meets PINNs.**   It is worth noting that in variational Monte Carlo applications, samples are drawn from a special probability density $x_i \sim p(x)$ where $p(x)$ depends on the wavefunction itself. However, this does not change the fundamental structure of the algorithms or the transferability of techniques from one domain to the other. The unifying principle is that in both application domains a single probability distribution is used to define the loss function, the natural gradient direction, and the sampling algorithm. The only difference is that in variational Monte Carlo the distribution happens to be dependent on the wavefunction, whereas for PINNs the distribution is a weighted mixture of the uniform distributions over the interior and the boundary.

**Sketching**   Randomized algorithms have made a major impact on numerical linear algebra over the last two decades, providing faster algorithms for many important problems including low-rank matrix approximations, linear systems, and eigenvalue solvers [13, 25, 29]. In the current work, we are particularly interested in the solution of regularized positive definite linear systems of the form

$$(A + \lambda I)x = y, \tag{3}$$

which are crucial for natural gradient methods (compare to equation 1). We will rely on the randomized Nyström approximation

$$\hat{A}_{\text{nys}} = (A\Omega)(\Omega^\top A\Omega)^\dagger (A\Omega)^\top \quad \text{where } \Omega \in \mathbb{R}^{n \times l} \text{ is standard normal,}$$

where $\dagger$ indicates the Moore-Penrose inverse and the rank $l$ can be chosen to control the cost and the accuracy of the approximation. This matrix provides the best positive semi-definite approximation of $A$ whose range coincides with the range of the sketch $A\Omega$; see [25] for a thorough discussion.

## 3   Methods

Let us get back to the PINN example problem in Section 2 and assume that $\mathcal{L}$ is a linear PDE operator[1] Define the residuals $r_\Omega(\theta)_i = 1/\sqrt{N_\Omega}(\mathcal{L}u_\theta(x_i) - f(x_i))$ with $i = 1, \ldots, N_\Omega$ and $r_{\partial\Omega}(\theta)_i = 1/\sqrt{N_{\partial\Omega}}(u_\theta(x_j^b) - g(x_j^b))$ with $j = 1, \ldots, N_{\partial\Omega}$. The loss function can then be written in a standard nonlinear least-squares form

$$L(\theta) = \frac{1}{2}\|r(\theta)\|^2, \quad \text{with} \quad r(\theta) = \begin{pmatrix} r_\Omega(\theta) \\ r_{\partial\Omega}(\theta) \end{pmatrix}$$

---

[1]If $\mathcal{L}$ is nonlinear, we use its linearization.

---

**Algorithm 1** SPRING for PINNs

---

**Require:** damping $\lambda$, momentum $\mu$, learning rate schedule $\eta_k$, initial guess $\theta_0$, norm constraint $C$
 1: **Initialize:**
 2:     $\theta \leftarrow \theta_0, \phi_0 \leftarrow 0$
 3: **for** $k = 1, \ldots, K$ **do**
 4:     Sample a new batch $(x_1, \ldots, x_{N_\Omega}, x_1^b, \ldots, x_{N_{\partial\Omega}}^b)$
 5:     Calculate $J_k, r_k$
 6:     $\zeta_k \leftarrow r_k - \mu J_k \phi_{k-1}$                                $\triangleright$ Residual shift for SPRING; see equations 7-8
 7:     $\phi_k \leftarrow J_k^\top \left( J_k J_k^\top + \lambda I \right)^{-1} \zeta_k$                    $\triangleright$ Woodbury form of ENGD, see equation 5
 8:     $\phi_k \leftarrow (\phi_k + \mu\phi_{k-1})/\sqrt{1 - \mu^{2k}}$              $\triangleright$ Add back the shift and bias correection
 9:     $\theta \leftarrow \theta - \phi_k \cdot \min(\eta_k, \sqrt{C}/||\phi_k||)$                          $\triangleright$ See Section 3.1 of [11]
10: **end for**
11: **return** Trained parameters $\theta$

---

and we set $N = N_\Omega + N_{\partial\Omega}$ and $r_k = r(\theta_k)$ and $J_k = J_\theta\, r(\theta_k)$. With this notation, it holds $G(\theta_k) = J_k^\top J_k$ and $\nabla L(\theta_k) = J_k^\top r_k$ and the natural gradient scheme equation 1 can be rewritten as

$$\theta_{k+1} = \theta_k - \eta_k(J_k^\top J_k + \lambda I)^{-1} J_k^\top r_k. \tag{4}$$

We again stress that, since $J_k$ is of shape $(N, P)$ the matrix $G(\theta_k)$ is of shape $(P, P)$, hence prohibitively large with a computational cost of $\mathcal{O}(P^3)$ for the linear solution.

**1) Leveraging Woodbury's Identity.** We now use the push-through identity, an intermediate step in the proof of the well-known Woodbury formula, to reduce the computational cost to $\mathcal{O}(N^2 P)$. While these are well-known matrix identities, their relevance for natural gradient methods in the context of quantum many-body problems was only recently realized in the VMC community and proposed as minimal norm stochastic reconfiguration (minSR) [4] and further refined by [36]. In our notation, the push-through identity is

$$\underbrace{(J_k^\top J_k + \lambda I)^{-1}}_{\in \mathbb{R}^{P \times P}} J_k^\top r_k = J_k^\top \underbrace{(J_k J_k^\top + \lambda I)^{-1}}_{\in \mathbb{R}^{N \times N}} r_k. \tag{5}$$

Using the right-hand side (ENGD-W) instead of the left, the *kernel* matrix $J_k J_k^\top \in \mathbb{R}^{N \times N}$ now lives in sample space and is thus cheap to invert. Indeed, the complexity $\mathcal{O}(N^2 P)$ of *computing* the kernel matrix now dominates the computational cost, a drastic improvement over the $O(P^3)$ cost of the original ENGD. The matrix $J_k J_k^\top$ is the neural tangent kernel [15], and various ways of efficiently computing it have been discussed in [30]. We discuss our approach in the implementation details in Section 4. We stress again that using the kernel form drastically improves the scalability of ENGD, allowing network and batch sizes up to $P = 10^6$ and $N = 10^3$ on consumer hardware while – up to floating point arithmetic – performing the exact scheme of equation 4. It is worth noting that the $O(N^2 P)$ complexity achieved by applying the Woodbury identity is still quadratic in the batch size $N$, whereas by applying further approximations K-FAC is able to achieve linear scaling with respect to the batch size. We have not observed this quadratic scaling to be a bottleneck for realistic batch sizes.

**2) Introducing Momentum.** The major drawback of the kernel formulation relative to methods that work with $G(\theta_k)$ directly[2] is that it makes it difficult to aggregate curvature information over multiple training iterations, which can lead to poor performance in highly stochastic settings. This problem was addressed by the introduction of the SPRING algorithm, which builds on MinSR by incorporating a special type of momentum that is tailored to accelerate the convergence of natural gradient methods [11]. To understand the SPRING algorithm, note first that the ENGD update of equation 5 is also the solution to a Tikhonov-regularized least-squares problem:

$$J_k^\top (J_k J_k^\top + \lambda I)^{-1} r_k = \arg\min_{\phi} ||J_k\phi - r_k||^2 + \lambda||\phi||^2. \tag{6}$$

SPRING modifies the regularization term to incorporate the previous momentum direction $\phi_{k-1}$:

$$\phi_k = \arg\min_{\phi} ||J_k\phi - r_k||^2 + \lambda||\phi - \mu\phi_{k-1}||^2. \tag{7}$$

---

[2]or approximations thereof, like K-FAC.

**Algorithm 2** GPU-Efficient Randomized Nyström Approximation

---

**Require:** $A \in \mathbb{S}^n_+(\mathbb{R})$, target rank $\ell$, regularizer $\lambda > 0$
**Ensure:** $(B, L)$ so that

$$\hat{A}_{\text{nys}} = BB^T, \qquad (\hat{A}_{\text{nys}} + \lambda I)^{-1}v = \frac{1}{\lambda}v - \frac{1}{\lambda}B\big(L^{-T}(L^{-1}(B^Tv))\big).$$

1: $\Omega \leftarrow \text{randn}(n, \ell)$            ▷ Gaussian test matrix
2: $Y \leftarrow A\,\Omega$
3: $\nu \leftarrow \text{eps}\big(\|Y\|_F\big)$            ▷ small shift
4: $Y_\nu \leftarrow Y + \nu\,\Omega$            ▷ embed shift in $A + \nu I$
5: $C \leftarrow \text{chol}\big(\Omega^T Y_\nu\big)$
6: $B \leftarrow Y_\nu\,C^{-1}$            ▷ Cholesky solve instead of inversion
7: $R \leftarrow B^T B + \lambda I$
8: $L \leftarrow \text{chol}(R)$            ▷ for Woodbury solve

---

This is justified by a connection to the randomized block Kaczmarz method and yields significant acceleration relative to MinSR [11]. This new optimization problem has closed-form solution

$$\phi_k = \mu\phi_{k-1} + J_k^\top(J_k J_k^\top + \lambda I)^{-1}(r_k - \mu J_k \phi_{k-1}), \tag{8}$$

and the cost of evaluating this formula is only negligibly higher than for MinSR. Note that MinSR is recovered by setting $\mu = 0$. Additional insights on the behavior of the iteration 8 and the function of the hyperparameters $\lambda, \mu$ can be found in [8, 12]. As a new contribution, we add a bias correction of $1/\sqrt{1 - \mu^{2t}}$ to the SPRING algorithm as is customarily done in other momentum-based optimizers [17]. The precise algorithm is given in Algorithm 1.

**3) Nyström on the Kernel & GPU-Efficient Implementation.** We introduce randomization into ENGD and SPRING by applying a randomized Nyström approximation of the $N \times N$ kernel matrix, potentially reducing the per-iteration cost even beyond the $O(N^2 P)$ cost attained via Woodbury's identity. Unfortunately, the well-known stable algorithm for applying the Nyström approximation (see [10], algorithm 2.1) requires an SVD of a $N \times S$ matrix. This cost is asymptotically negligible but nonetheless we have found that it is very significant in practice because the SVD runs extremely slowly on GPU. To ameliorate this problem, we propose a GPU-efficient Nyström approximation which we summarize in Algorithm 2. Our proposal runs an order of magnitude faster on GPU (see Appendix B) and enables speed-ups to be obtained even when $S$ is only a single order of magnitude smaller than $N$.

The main differences relative to the standard stable algorithm are 1) we skip the QR decomposition of the test matrix $\Omega$ since random Gaussian matrices are already likely to be well-conditioned, 2) we skip the SVD step and, as a result, we technically return a Nyström approximation of $A + \nu I$ for a very small value of $\nu$, and 3) we apply the Woodbury matrix identity (again) to reduce the cost of matrix-vector products with the inverse of the Nyström approximation.

We leverage this GPU-efficient randomized Nyström approximation in a sketch-and-solve framework [38]; for example for ENGD the randomized version uses the approximation

$$J_k^\top(J_k J_k^\top + \lambda I)^{-1}r_k \approx J_k^\top\big(\text{nys}(J_k J_k^\top) + \lambda I\big)^{-1}r_k, \tag{9}$$

with the linear solve implemented following algorithm 2. It is known that such a sketch-and-solve approach can require very large sketch sizes to achieve high accuracy solutions, which has inspired the proposal of an alternative sketch-and-precondition approach which can attain arbitrary accuracies with any sketch size [10]. However, in the PINN setting the CG iterations required by such an approach involve additional differentiation passes through the operator $\mathcal{L}$, which we have found nullifies any performance benefit from randomization in practice.

**4) Tracking the effective dimension.** The accuracy of the randomized Nyström approximation depends on the effective dimension [10] of the regularized matrix $A + \lambda I$. The effective dimension measures the degrees of freedom of the problem after regularization and serves as a smooth quantifier of the number of eigenvalues larger than $\lambda$. The smaller the effective dimension, the smaller the sketch size needed to form an accurate Nyström approximation and the better performance we can

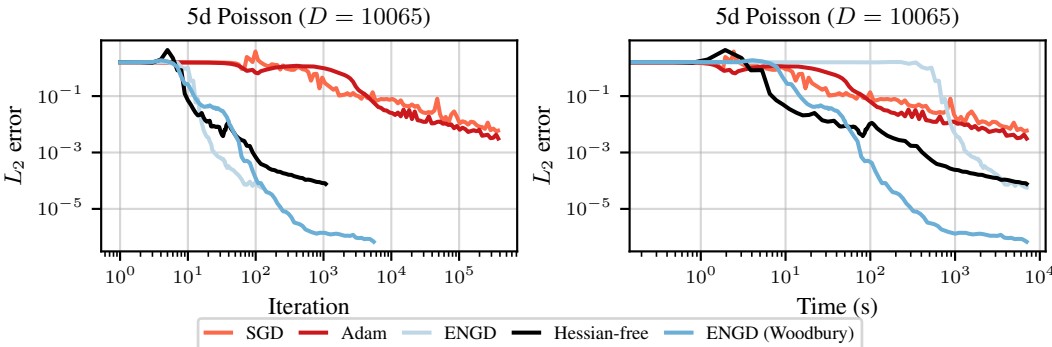

Figure 2: **5D Poisson:** Performance comparison of different optimization algorithms on a five-dimensional Poisson problem discretized with $10\,065$ parameters, trained with a tanh-activated multilayer perceptron. Introducing the Woodbury matrix identity allows ENGD to take more than 30 times more steps and outperform the Hessian-free optimizer by a significant margin.

expect from our randomized algorithms. The effective dimension is defined as

$$d_{\text{eff}}(A) = \text{Tr}(A(A + \lambda I)^{-1}) = \sum_{i=1}^{n} \frac{\lambda_i}{\lambda_i + \lambda}.$$

## 4   Experiments

To show the implications of our contributions, we present 4 experiments, each consisting of more than 1200 runs on average. First, we show that using the kernel matrix improves ENGD beyond the performance of the usual baselines. Second, we show that incorporating momentum using SPRING further accelerates convergence and achieves state-of-the-art results. Moreover, SPRING removes the need for an expensive line search and performs particularly well in high dimensional settings. Third, we show that our GPU-efficient Nyström approximation can accelerate the early phases of training for low-dimensional problems with large batch sizes; in most cases, however, it appears preferable to use the exact Woodbury formula. Fourth, we show that the regularized kernel matrix has an effective dimension that is only mildly smaller than the batch size $N$, which explains the limited benefits achieved by randomization. Importantly, our randomized algorithms still far outperform the original ENGD method, and it is only when comparing to the new Woodbury variants that they lose their appeal.

**Setup**   To demonstrate our methods, we consider a Poisson equation $-\Delta u(x) = f(x)$ with different right-hand sides and boundary conditions on the unit square $x \in [0, 1]^d$ for $d = 5, 100$. For $d = 5$, we use as manufactured solution $u(x) = \sum_{i=1}^{5} \cos(\pi x_i)$ and right-hand side $f = \pi^2 u$. For $d = 100$, we use $u(x) = ||x||_2^2$ for $x \in \partial[0, 1]^{100}$ and consequently $f = -2d$. Furthermore, we consider a 4+1d Heat equation $\partial_t u(t, x) - \kappa \Delta_x u(t, x) = 0$ for $t \in [0, 1], x \in [0, 1]^4$ and boundary conditions $u(0, x) = \sum_{i=1}^{4} \sin(2x_i)$ and $u(t, x) = \exp(-t) \sum_{i=1}^{4} \sin(2x_i)$, this last one with $x \in \partial[0, 1]^4$. At last, we consider a 9+1d Fokker-Planck equation in logarithmic space given by

$$\partial_t q(t, x) - \frac{d}{2} - \frac{1}{2} \nabla q(t, x) \cdot x - ||\nabla q(t, x)||^2 - \Delta q(t, x) = 0, \quad q(0) = \log(p^*(0)),$$

where $d = 9, t \in [0, 1]$ and $x \in \mathbb{R}^9$. In practice, $x \in [-5, 5]^9$. The solution $q^*$ is given by $q^* = \log(p^*)$ with $p^*(t, x) \sim \mathcal{N}(0, \exp(-t)I + (1 - \exp(-t))2I)$. For further results see Appendix A.

**Implementation**   We use the same sequential architecture consisting of four hidden layers and a single output for all problems, varying the width of the layers as a function of $d$. Moreover, we implement the model using Taylor-mode forward differentiation [2], and use Jacobian vector products to optimize kernel creation and kernel vector products [15]. To implement a matrix-free alternative to ENGD, we adopt the Hessian-free optimization method [23, 43], which applies truncated conjugate gradient iterations and computes exact Gramian-vector products to improve gradient conditioning.

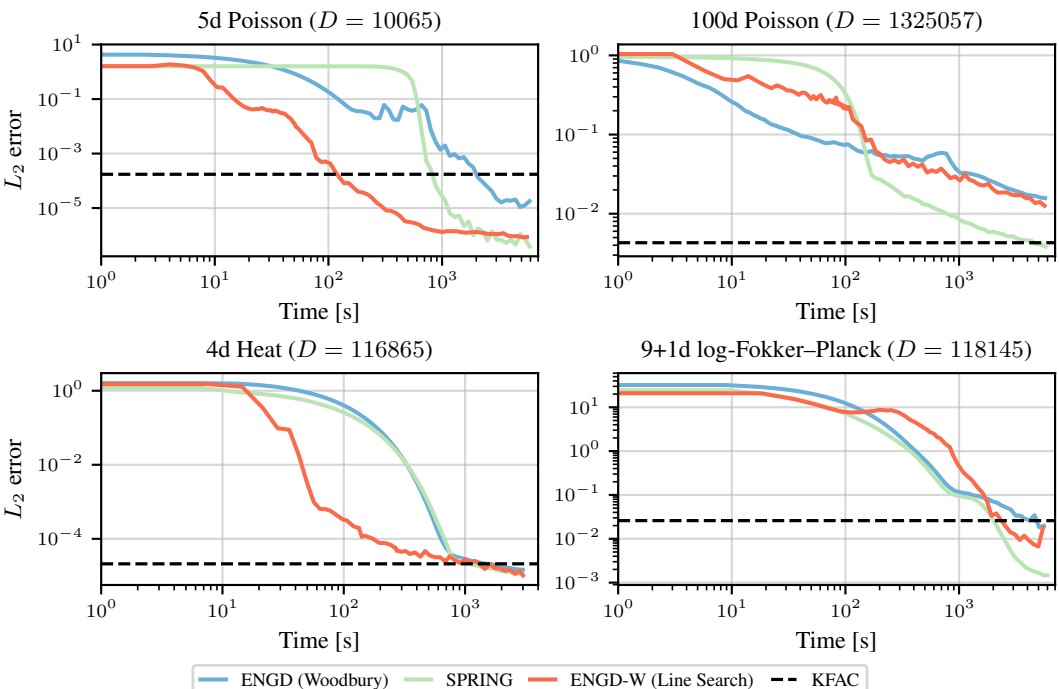

Figure 3: **Performance of ENGD and SPRING on four problems: 5d and 100d Poisson, 4+1d Heat, and 9+1d log-Fokker-Plancks.** SPRING achieves $L^2$ errors not previously seen in the high dimensional settings, the 100d Poisson and 9+1d log-Fokker-Planck problems. In the latter, the error is an order of magnitude lower that the previous state-of-the-art KFAC.

We evaluate two other baselines: SGD with tuned momentum and learning rate, and Adam with a tuned learning rate. Hyperparameter tuning is performed using Weights & Biases [47], as detailed in Appendix A. Our tuning strategy involves an initial exploratory stage with 50 trials over broad parameter ranges, followed by a refinement stage with another 50 trials in narrowed regions of interest. Performance is evaluated based on the lowest $L^2$-error in a validation set with a known PDE solution. All experiments are run on a uniform hardware setup, an RTX 6000 GPU cluster (24 GiB memory) using double precision, and each optimizer is given an equal compute time budget on the same fixed PINN task. The complete ranges of hyperparameters, the selected configurations and the dynamics of training in terms of the iteration count are available in Appendix A.

**1) Introducing Woodbury into natural gradient methods for PINNs.** We first reproduce the 5-dimensional Poisson problem of Dangel, Müller, and Zeinhofer [6], with batch size of 3500. We compare the baselines against ENGD-W (see Figure 2), and verify that the exact ENGD already attains top-tier solution accuracy. Incorporating the Woodbury identity then increases the iteration speed by more than a factor of 30 and surpasses all other approaches, including the original ENGD and Hessian-free optimization, in both efficiency and accuracy.

> **Woodbury is the *right* way to implement second-order methods such as ENGD.**

**2) Incorporating momentum to further accelerate ENGD.** We next incorporate momentum using the SPRING algorithm. Figure 3 shows that momentum increases the final accuracy in all cases, with a large benefit in the high-dimensional setting, i.e. the 100D Poisson problem and the 9+1d log-Fokker-Planck problem. It is notable that the SPRING algorithm does not use a line search and is able to outperform ENGD both with and without its line search. The final accuracy attained by both ENGD with Woodbury and SPRING surpasses even that attained by [6] using the more complicated KFAC algorithm. This is accentuated with SPRING attaining an $L^2$-error an order of magnitude lower than KFAC in the 9+1d log-Fokker-Planck problem.

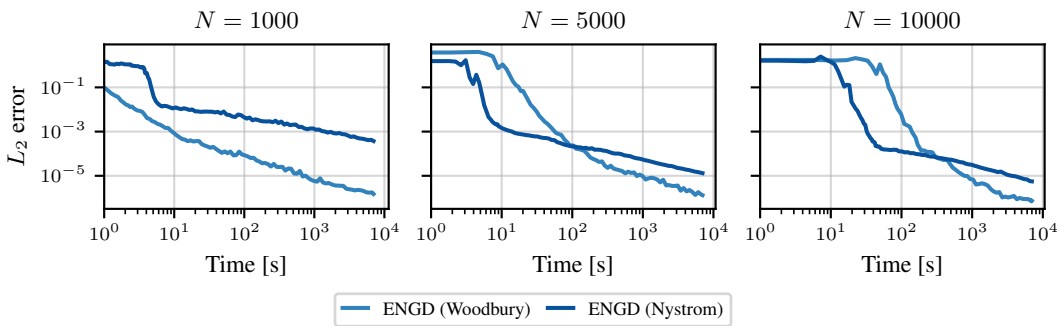

Figure 4: **Effect of Nyström on ENGD for the 5D Poisson equation.**



**SPRING yields state-of-the-art performance without the complexity of KFAC.**



**3) Randomization for large batch sizes.** We have shown that adding momentum to ENGD-W with a tuned learning rate is the best strategy to optimize PINNs. When the batch size is very large, even the computation of the kernel matrix can be quite expensive, motivating the use of randomized algorithms to reduce the cost. We explore the use of a randomized Nyström approximation to accelerate the computation and inversion of the regularized kernel matrix $J_k J_k^\top + \lambda I$. In the first set of experiments, shown in Figure 4, we compare ENGD-W with two randomized variants, one using the standard stable Nyström approximation and the other using our new GPU-efficient variant, Algorithm 2. We test batch sizes of $1000$, $10\,000$, and $50\,000$, all under our standard line-search procedure and all with a sketch size of 10% of the batch size. The 10% batch size is chosen because larger batch sizes, even the GPU-efficient algorithm cannot provide substantial cost reductions. We find that randomization can accelerate the early phase of the training, especially when the batch size is large, but ultimately exact computations are needed to reach high accuracy results.

In the second set of experiments, shown in Figure 5, we compare SPRING against its two corresponding randomized variants on a 100D Poisson problem. In this case, the performance of the randomized algorithms is approximately equal to or worse than the performance of SPRING with exact computations. The main reason that we do not observe any speed-ups from randomization here is that for high-dimensional problems, the cost of the training is increasingly dominated by the cost of differentiating through the operator $\mathcal{L}$, which scales linearly with the problem dimension. Thus, the acceleration of the kernel computation and inversion becomes less relevant. For the 5d problem, we find a speedup of $2\times$ for a sketch size of 10% of $N$, and no speedup for values above 25% of $N$. Given the cost of the matvecs and the Laplacian operator, this is to be expected.



**Randomization accelerates the early phases of training for low-dimensional problems with large batches. It remains an open problem to extend this acceleration to broader scenarios.**



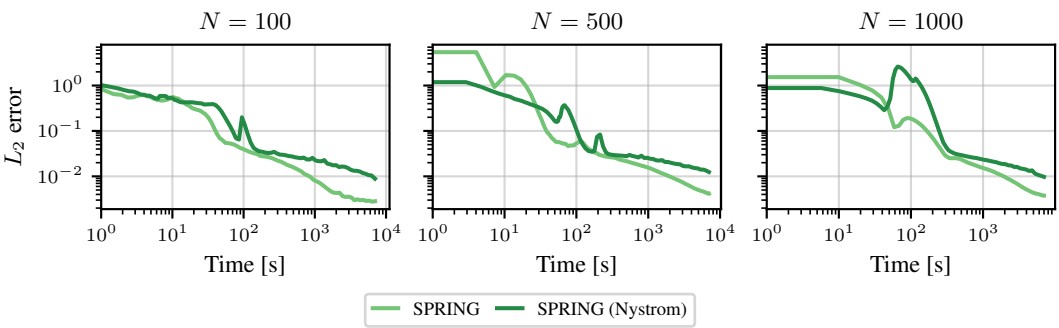

Figure 5: **Effect of Nyström on SPRING for the 100D Poisson equation.**

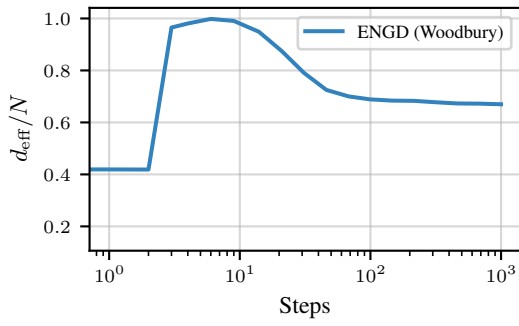
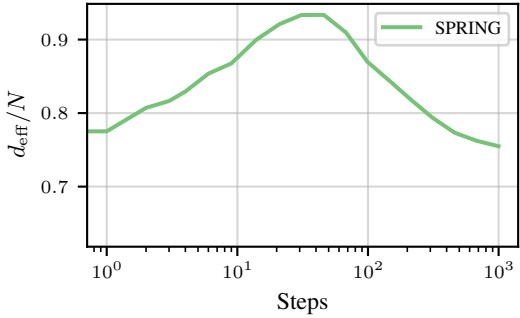

(a) **Effective dimension of ENGD's regularized kernel matrix in the 5D Poisson experiment relative to the batch size.** $N = 3500$

(b) **Effective dimensions of SPRING's regularized kernel matrix in the 100D Poisson experiment relative to the batch size.** $N = 150$

**4) Effective dimension of the regularized kernel matrix.** To understand the limited benefits of randomization, we track the effective dimension of the kernel matrix for both ENGD-W in the 5D problem and SPRING in the 100D problem, with results in Figure 6a and Figure 6b, respectively. The effective dimension plateaus at more than 50% of the batch size, which explains why randomization with a sketch size of 10% results in a loss of accuracy. Unfortunately, the gap between the effective dimension and the batch size $N$ is not large enough to squeeze out significant performance gains without losing accuracy.

> **Randomization suffers because the sketch sizes needed to attain significant cost savings are too small to accurately approximate the kernel matrix.**

Our findings contrast those of McKay et al. [26] which reported benefits from using randomization in original version of ENGD. We compare our randomized algorithms to the much stronger baseline of ENGD-W, and it is exactly the introduction of the Woodbury identity that nullifies the benefits of randomization by shifting the problem from the high dimensional parameter space to the much lower dimensional sample space, leaving little room for further dimensionality reduction.

## 5   Conclusion

We have *significantly* improved energy natural gradient descent for PINNs by: 1) Woodbury's identity to accelerate the inversion of the kernel matrix, thus slashing per-iteration complexity, 2) SPRING, a momentum scheme that accelerates the convergence of natural gradient methods, and 3) a GPU-efficient Nyström sketch-and-solve approach to approximate the kernel matrix for further cost reductions in some settings. We demonstrate our methods across four settings: 5d and 100d Poisson equations, 4+1d Heat equation and 9+1d log-Fokker-Planck equation with batch sizes up to 10 000. In the 5d Poisson case, our methods achieve the same sub-$10^{-3}$ $L^2$-error as ENGD up to 75× faster, while in the 100d case, SPRING outperforms all previously seen optimizers. In the 4+1d Heat case our methods perform competitively with previous state of the art methods like KFAC, and in the 9+1d log-Fokker-Planck problem, SPRING achieves $L^2$-errors an order of magnitude lower than previous methods. Regarding randomization, we find promising results for low-dimensional problems with large batch sizes, and we identify barriers to achieving acceleration more generally.

**Limitations & future directions**   Despite promising results in low-dimensional, large batch settings, our randomized approaches exhibit diminishing returns in the late optimization stages and fail to scale to high-dimensional problems. Future works can explore alternative approaches to randomization, but will need to contend with the peculiarities of the PINN setting in which matrix-vector products with the kernel are very expensive. Moreover, our current analysis focuses on fixed Nyström rank, leaving open questions about how sketch dimension and adaptive rank selection affect performance across diverse tasks. Future work should also explore how to adaptively set the hyperparameters of the ENGD-W and SPRING algorithms to yield fast, black-box optimizers.

## Acknowledgments and Disclosure of Funding

M.Z. acknowledges support from an ETH Postdoctoral Fellowship for the project "Reliable, Efficient, and Scalable Methods for Scientific Machine Learning". G.G. acknowledges support from the U.S. Department of Energy, Office of Science, Office of Advanced Scientific Computing Research, Department of Energy Computational Science Graduate Fellowship under Award Number DE-SC0023112. A.G-C thanks Jeffrey Ren for helpful observations on the optimizer's norm-constraint setting used in plot generation. Resources used in preparing this research were provided, in part, by the Province of Ontario, the Government of Canada through CIFAR, and companies sponsoring the Vector Institute.

**Disclaimer:** This report was prepared as an account of work sponsored by an agency of the United States Government. Neither the United States Government nor any agency thereof, nor any of their employees, makes any warranty, express or implied, or assumes any legal liability or responsibility for the accuracy, completeness, or usefulness of any information, apparatus, product, or process disclosed, or represents that its use would not infringe privately owned rights. Reference herein to any specific commercial product, process, or service by trade name, trademark, manufacturer, or otherwise does not necessarily constitute or imply its endorsement, recommendation, or favoring by the United States Government or any agency thereof. The views and opinions of authors expressed herein do not necessarily state or reflect those of the United States Government or any agency thereof.

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

# A Experimental Details and Additional Results

## A.1 Hyper-Parameter Tuning Protocol

We tune the following optimizer hyper-parameters and otherwise use the PyTorch default values:

- **SGD:** learning rate, momentum
- **Adam:** learning rate
- **Hessian-free:** type of curvature matrix (Hessian or GGN), damping, whether to adapt damping over time (yes or no), maximum number of CG iterations
- **ENGD:** damping, factor of the exponential moving average applied to the Gramian, initialization of the Gramian (zero or identity matrix)
- **ENGD (Woodbury):** damping, learning rate (when fixed)
- **SPRING:** damping, momentum, learning rate (when fixed)
- **Randomized:** damping, learning rate (when fixed), sketch size

We use random search from Weights & Biases to determine the hyper-parameters, where run is executed in double precision and allowed to run for a given time budget, and we rank runs by the final $L^2$ error on a fixed evaluation data set. To compare, the hardware used was RTX 6000 GPUs with 24 GiB of RAM. The first state consists of a relatively wide search space and limit to 50 runs. The second stage is narrowed down to a smaller space based on the first stage, then re-run for another 50 runs. We will release the details of all hyper-parameter search spaces, as well as the hyper-parameters for the best runs in our implementation.

## A.2 5d Poisson Equation

**Setup** We consider a five-dimensional Poisson equation $-\Delta u(\boldsymbol{x}) = \pi^2 \sum_{i=1}^{5} \cos(\pi x_i)$ on the five-dimensional unit square $\boldsymbol{x} \in [0,1]^5$ with cosine sum right-hand side and boundary conditions $u(\boldsymbol{x}) = \sum_{i=1}^{5} \cos(\pi x_i)$ for $\boldsymbol{x} \in \partial[0,1]^5$. We sample training batches of size $N_\Omega = 3000, N_{\partial\Omega} = 500$ and evaluate the $L^2$ error on a separate set of $30\,000$ data points using the known solution $u_\star(\boldsymbol{x}) = \sum_{i=1}^{5} \cos(\pi x_i)$. All optimizers sample a new training batch each iteration, and each run is limited to 7000s. We use an MLP five-layer architecture whose linear layers are Tanh-activated except for the final one: $5 \rightarrow 64 \rightarrow 64 \rightarrow 48 \rightarrow 48 \rightarrow 1$ MLP with $D = 10\,065$ trainable parameters. ENGD-W and SPRING here make use of the inherited ENGD line search. Figure 7 visualizes the results.

**Best run details** The runs shown in Figure 7 correspond to the following hyper-parameters:

- **SGD:** learning rate: $2.895\,360 \times 10^{-3}$, momentum: $3 \times 10^{-1}$
- **Adam:** learning rate: $2.808\,451 \times 10^{-4}$
- **Hessian-free:** curvature matrix: GGN, initial damping: $1 \times 10^{-1}$, constant damping: no, maximum CG iterations: 350
- **ENGD:** damping: $1 \times 10^{-8}$, exponential moving average: 0, initialize Gramian to identity: yes
- **ENGD-W:** damping: $3.173\,212 \times 10^{-12}$
- **SPRING:** damping: $2.086\,287 \times 10^{-10}$; momentum: $3.115\,42 \times 10^{-1}$

**Search space details** The runs shown in Figure 7 were determined to be the best via a search with approximately 50 runs on the following search spaces which were obtained by refining an initially wider search ($\mathcal{U}$ denotes a uniform, and $\mathcal{LU}$ a log-uniform distribution):

- **SGD:** learning rate: $\mathcal{LU}([1 \times 10^{-3}; 1 \times 10^{-2}])$, momentum: $\mathcal{U}(\{0, 3 \times 10^{-1}, 6 \times 10^{-1}, 9 \times 10^{-1}\})$
- **Adam:** learning rate: $\mathcal{LU}([1 \times 10^{-4}; 5 \times 10^{-1}])$

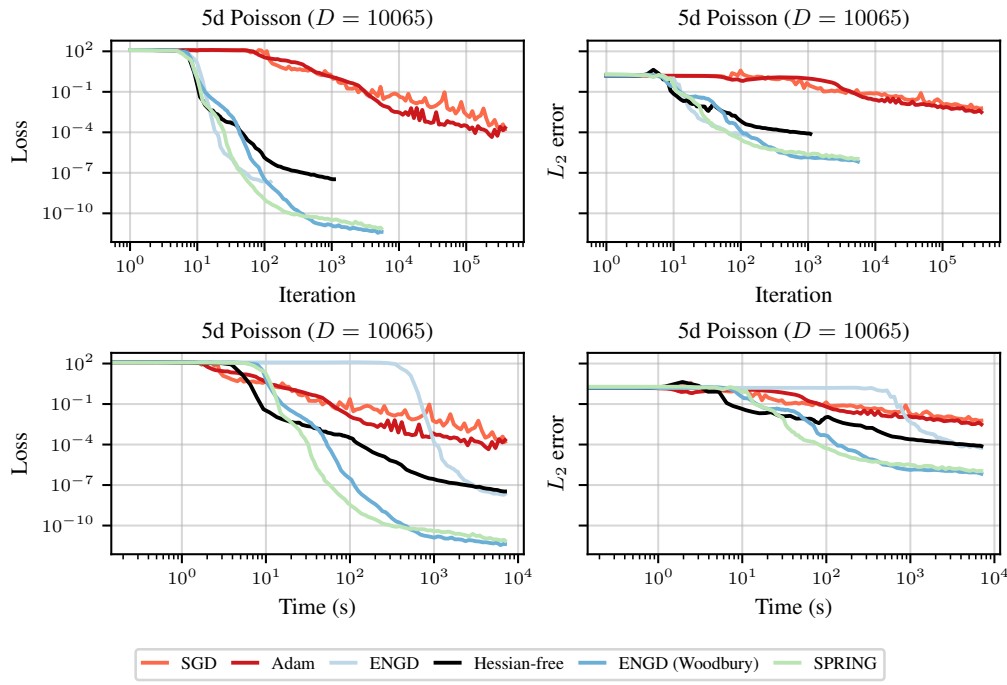

Figure 7: Training loss and evaluation $L^2$ error for learning the solution to a 5d Poisson equation over time and steps.

- **Hessian-free:** curvature matrix: $\mathcal{U}(\{\text{GGN}\})$, initial damping: $\mathcal{U}(\{100, 50, 10, 5, 1, 5 \times 10^{-1}, 1 \times 10^{-1}, 5 \times 10^{-2}\})$, constant damping: $\mathcal{U}(\{\text{no}\})$, maximum CG iterations: $\mathcal{U}(\{100, 150, 200, 250, 300, 350\})$
- **ENGD:** damping: $\mathcal{U}(\{1 \times 10^{-8}, 1 \times 10^{-9}, 1 \times 10^{-10}, 1 \times 10^{-11}, 1 \times 10^{-12}, 0\})$, exponential moving average: $\mathcal{U}(\{0, 3 \times 10^{-1}, 6 \times 10^{-1}, 9 \times 10^{-1}\})$, initialize Gramian to identity: $\mathcal{U}(\{\text{no}, \text{yes}\})$
- **ENGD-W:** damping: $\mathcal{LU}([1 \times 10^{-7}; 1])$
- **SPRING:** damping: $\mathcal{LU}([1 \times 10^{-10}; 1 \times 10^{-3}])$ momentum: $\mathcal{U}([0.0; 0.999])$

### A.2.1 Fixed learning rate

We repeat the previous experiments, now adding a search space of $\mathcal{LU}([1 \times 10^{-1}; 1 \times 10^{-4}])$ for the learning rate, see Figure 8. We note that this change meant adjusting the search space for SPRING's momentum to $\mathcal{LU}([0.8; 0.999])$. We find that the best parameters are:

- **ENGD-W:** damping: $6.804\,474 \times 10^{-8}$; learning rate: $5.2289 \times 10^{-2}$
- **SPRING:** damping: $6.811\,585 \times 10^{-10}$; momentum: $8.269\,66 \times 10^{-1}$; learning rate: $6.3502 \times 10^{-2}$

### A.2.2 Large batches

We now repeat the experiment using batch sizes of $1000$, $5000$, and $10\,000$ with the line search, to test the randomized approach, setting the sketch size to 10% of $N$. We can visualize the results in Figure 9.

### A.3 10d Poisson Equation with line search

**Setup** We consider a 10-dimensional Poisson equation $-\Delta u(\boldsymbol{x}) = 0$ on the 10-dimensional unit square $\boldsymbol{x} \in [0, 1]^5$ with zero right-hand side and harmonic mixed second order polynomial boundary conditions $u(\boldsymbol{x}) = \sum_{i=1}^{d/2} x_{2i-1} x_{2i}$ for $\boldsymbol{x} \in \partial[0, 1]^d$. We sample training batches of size $N_\Omega =$

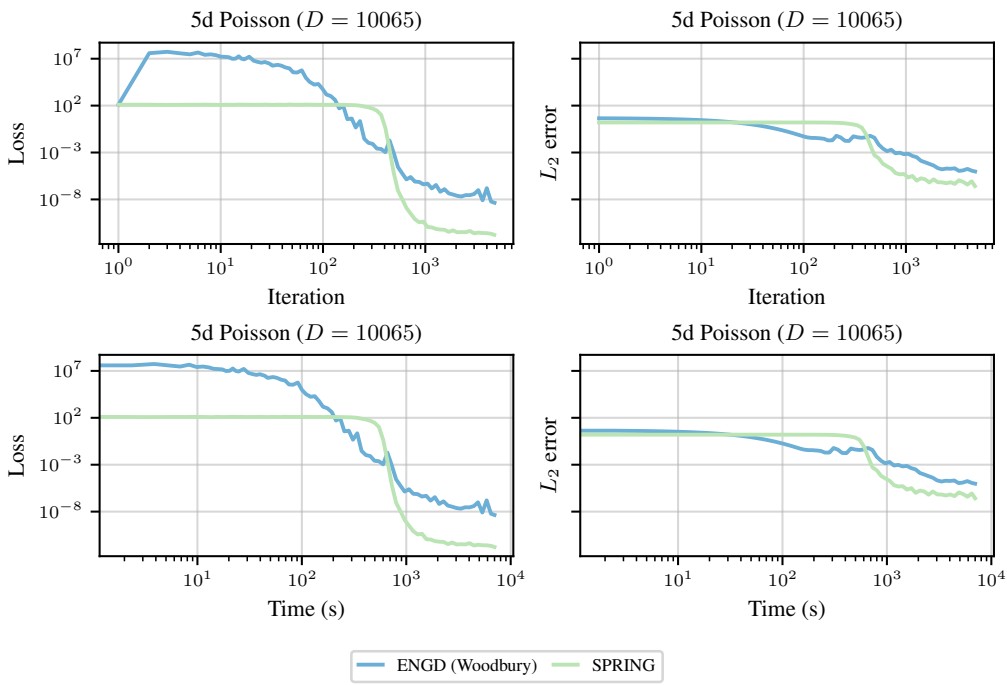

Figure 8: Training loss and evaluation $L^2$ error for learning the solution to a 5d Poisson equation over time and steps with fixed learning rate.

$3000$, $N_{\partial\Omega} = 1000$ and evaluate the $L^2$ error on a separate set of $30\,000$ data points using the known solution $u_\star(\boldsymbol{x}) = \sum_{i=1}^{d/2} x_{2i-1} x_{2i}$. Both optimizers sample a new training batch each iteration, and each run is limited to $7000\,\text{s}$. We use a $10 \rightarrow 256 \rightarrow 256 \rightarrow 128 \rightarrow 128 \rightarrow 1$ MLP with $D = 118\,145$ MLP whose linear layers are Tanh-activated except for the final one. Given the poor performance of SGD, Adam, and the Hessian-free, we no longer run them in this more complicated problem. Furthermore, the traditional ENGD runs out of memory for networks of this size. Figure 11 visualizes the results.

**Best run details**  The runs shown in Figure 11 correspond to the following hyper-parameters:

- **ENGD-W:** damping: $3.9 \times 10^{-7}$
- **SPRING:** damping: $1.7 \times 10^{-7}$ momentum: $9.053\,28 \times 10^{-1}$

**Search space details**  The runs shown in Figure 11 were determined to be the best via a random search on the following search spaces which each optimizer given approximately the same total computational time ($\mathcal{U}$ denotes a uniform, and $\mathcal{LU}$ a log-uniform distribution):

- **ENGD-W:** damping: $\mathcal{LU}([1 \times 10^{-7}; 1])$
- **SPRING:** damping: $\mathcal{LU}([1 \times 10^{-10}; 1 \times 10^{-3}])$; momentum: $\mathcal{LU}([0.6; 0.999])$

### A.3.1 Fixed learning rate

We repeat the previous experiments, now adding a search space of $\mathcal{LU}([1 \times 10^{-1}; 1 \times 10^{-4}])$ for the learning rate, see Figure 12. We find that the best parameters are:

- **ENGD-W:** damping: $1.579 \times 10^{-5}$; learning rate: $8.7024 \times 10^{-2}$
- **SPRING:** damping: $4.98 \times 10^{-5}$; momentum: $9.6764 \times 10^{-1}$; learning rate: $6.034\,67 \times 10^{-2}$

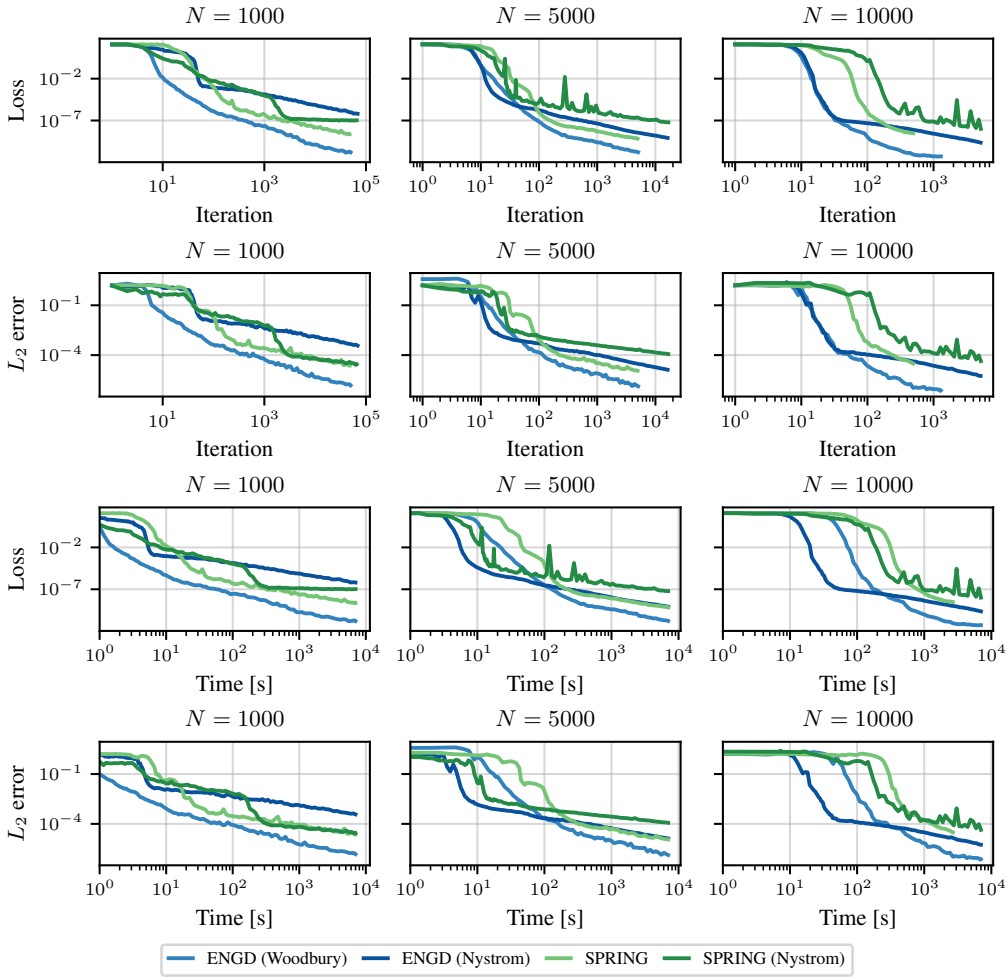

Figure 9: Training loss and evaluation $L^2$ error for learning the solution of ENGD-W to a 5d Poisson equation over time and steps with large batch sizes and randomization.

## A.4  100-d Poisson Equation with line search

**Setup**  Here, we consider a 100d Poisson equations $-\Delta u(\boldsymbol{x}) = f(\boldsymbol{x})$ with zero right-hand side $f(\boldsymbol{x}) = 0$, harmonic mixed second order polynomial boundary conditions $u(\boldsymbol{x}) = \sum_{i=1}^{d/2} x_{2i-1} x_{2i}$ for $\boldsymbol{x} \in \partial[0,1]^d$, and known solution $u_\star(\boldsymbol{x}) = \sum_{i=1}^{d/2} x_{2i-1} x_{2i}$. We assign each run a budget of $10\,000\,\mathrm{s}$. We tune the optimizer-hyperparameters described in Appendix A.1 using random search. We use a $100 \to 768 \to 768 \to 512 \to 512 \to 1$ MLP with $D = 1\,325\,057$ MLP whose linear layers are Tanh-activated except for the final one. This architecture is again too large for ENGD to optimize. Figure 13 visualizes the results.

**Best run details**  The runs shown in Figure 13 correspond to the following hyper-parameters:

- **ENGD-W:** damping: $4.7772 \times 10^{-3}$
- **SPRING:** damping: $3.0106 \times 10^{-2}$ momentum: $6.763\,35 \times 10^{-1}$

**Search space details**  The runs shown in Figure 13 were determined to be the best via a random search on the following search spaces which each optimizer given approximately the same total computational time ($\mathcal{U}$ denotes a uniform, and $\mathcal{LU}$ a log-uniform distribution):

- **ENGD-W:** damping: $\mathcal{LU}([1 \times 10^{-10}; 1 \times 10^{-1}])$
- **SPRING:** damping: $\mathcal{LU}([1 \times 10^{-10}; 1 \times 10^{-3}])$; momentum: $\mathcal{LU}([0.6; 0.999])$

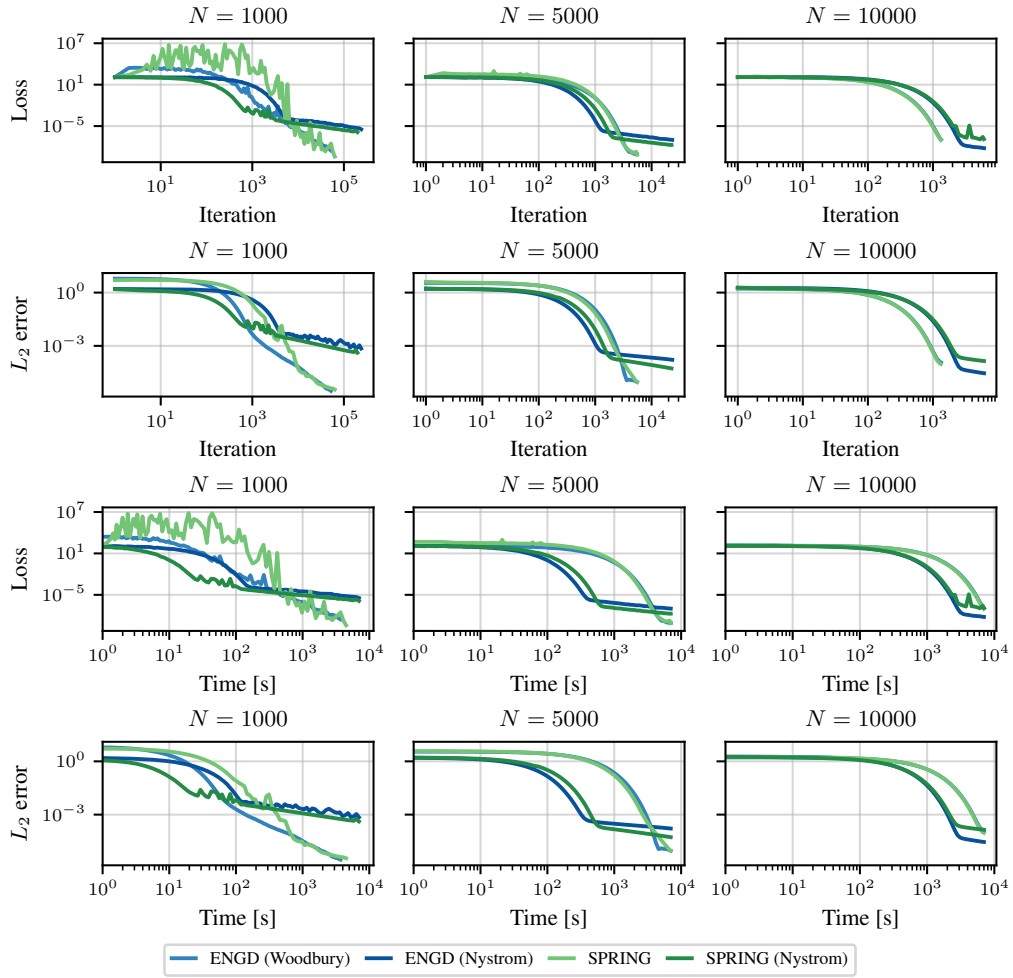

Figure 10: Training loss and evaluation $L^2$ error for learning the solution to a 5d Poisson equation over time and steps with large batch sized and fixed learning rate.

### A.4.1 Fixed learning rate

We repeat the previous experiments, now adding a search space of $\mathcal{LU}([1 \times 10^{-1}; 1 \times 10^{-4}])$ for the learning rate, see Figure 12. We find that the best parameters are:

- **ENGD-W:** damping: $6.233 \times 10^{-7}$; learning rate: $9.118 \times 10^{-2}$
- **SPRING:** damping: $3.0116 \times 10^{-2}$; momentum: $9.8386 \times 10^{-1}$; learning rate: $9.2362 \times 10^{-2}$

### A.4.2 Large batches

We now repeat the experiment using batch sizes of 1000, 5000, and 10 000 with the line search, in order to test the randomized approach, setting the sketch size to 10% of $N$ and fixed the learning rate with the previously introduced search space. We can visualize the results in Figure 9.

### A.5 4+1d Heat equation

**Setup** We consider a 4+1-dimensional heat equation $\partial_t u(t, x) - \kappa \Delta_x u(t, x) = 0$ with $\kappa = \frac{1}{4}$ on the four-dimensional unit square and unit time interval, $x, t \in [0, 1]^4 \times [0, 1]$. The equation has spatial boundary conditions $u(t, x) = \exp(-t) \sum_{i=1}^{4} \sin(2x_i)$ for $t, x \in [0, 1] \times \partial[0, 1]^4$ throughout time, and initial value conditions $u(0, x) = \sum_{i=1}^{4} \sin(2x_i)$ for $x \in [0, 1]^4$. We sample training

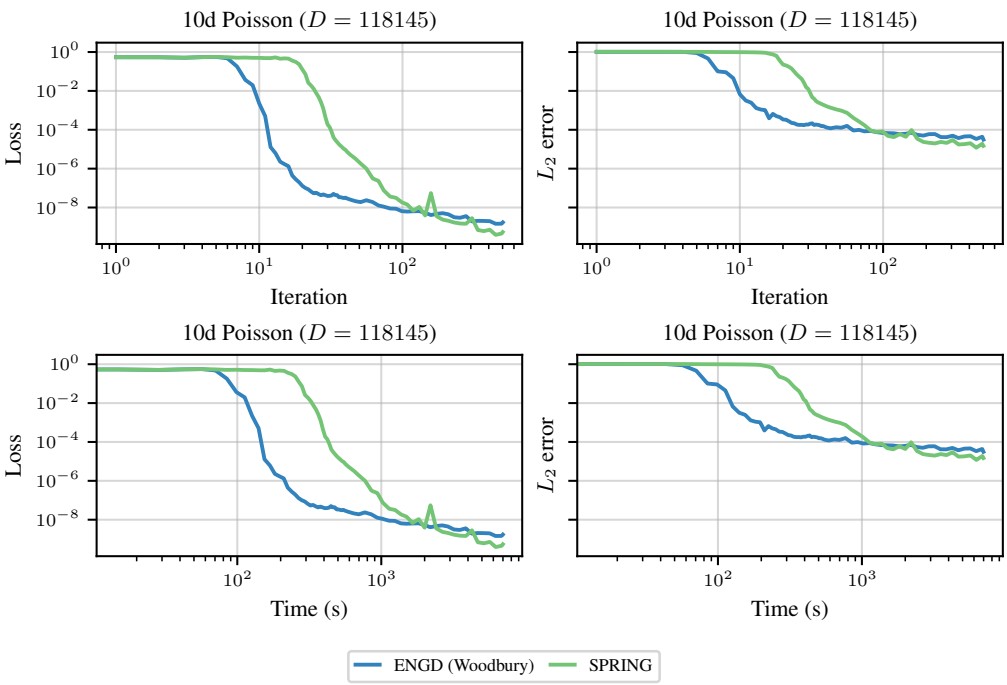

Figure 11: Training loss and evaluation $L^2$ error for learning the solution to a 10d Poisson equation over time and steps.

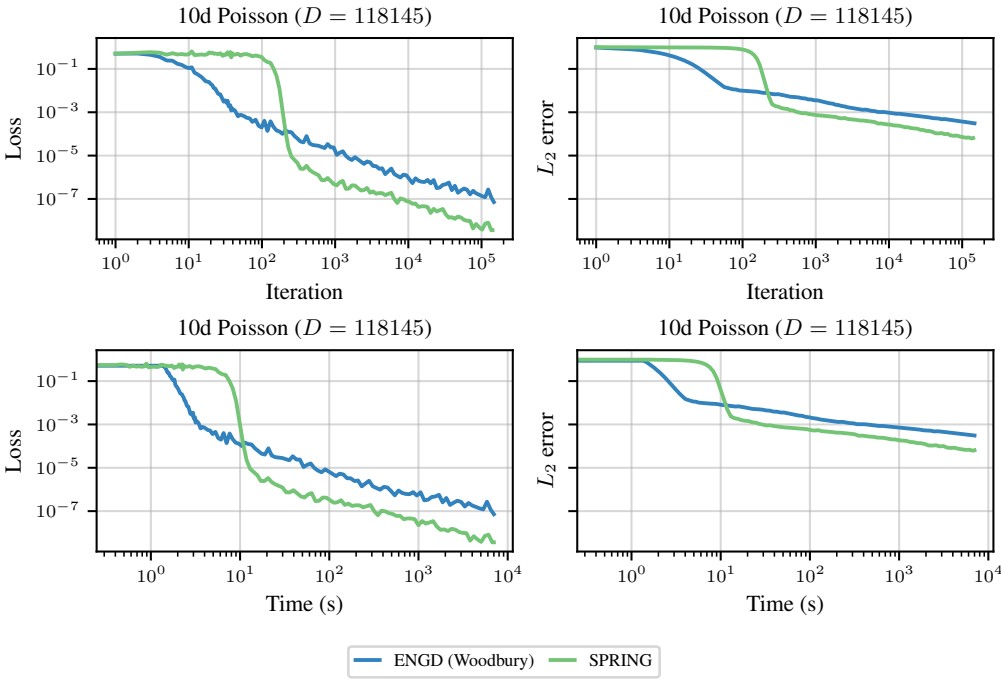

Figure 12: Training loss and evaluation $L^2$ error for learning the solution to a 10d Poisson equation over time and steps with fixed learning rate.

batches of size $N_\Omega = 3000, N_{\partial\Omega} = 500$ ($N_{\partial\Omega/2}$ points for the initial value and spatial boundary conditions each) and evaluate the $L^2$-error on a separate set of 30000 data points using the known solution $u_\star(t, x) = \exp(-t) \sum_{i=1}^{4} \sin(2x_i)$. We sample a new training batch each iteration. Each

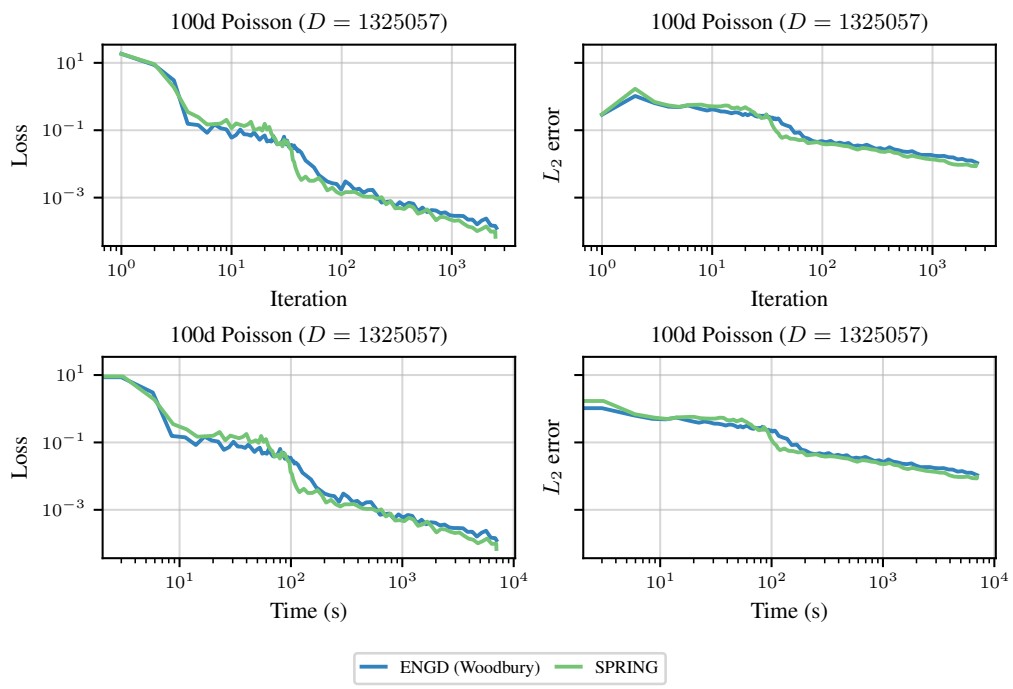

Figure 13: Training loss and evaluation $L^2$ error for learning the solution to high-dimensional Poisson equations over time and steps using random search.

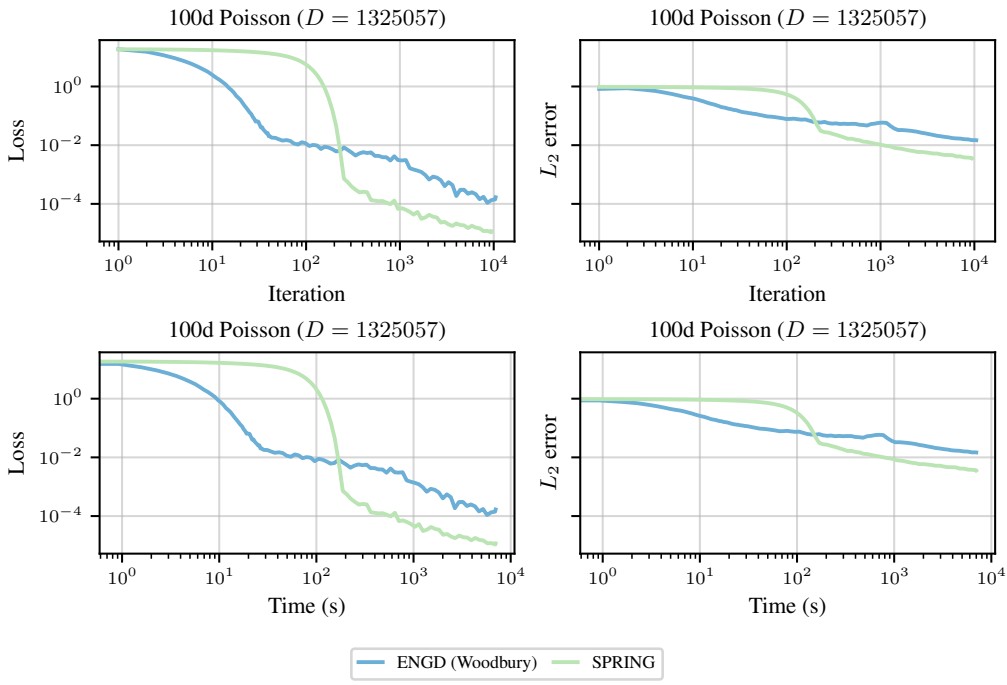

Figure 14: Training loss and evaluation $L^2$ error for learning the solution to a 100d Poisson equation over time and steps with fixed learning rate.

run is limited to 3000 s. We use a five-layer MLP architecture whose linear layers are Tanh-activated except for the final one: $5 \to 256 \to 256 \to 128 \to 128 \to 1$ with $D = 116864$ trainable weights.

**Best run details** The runs shown in Figure 16 correspond to the following hyper-parameters:

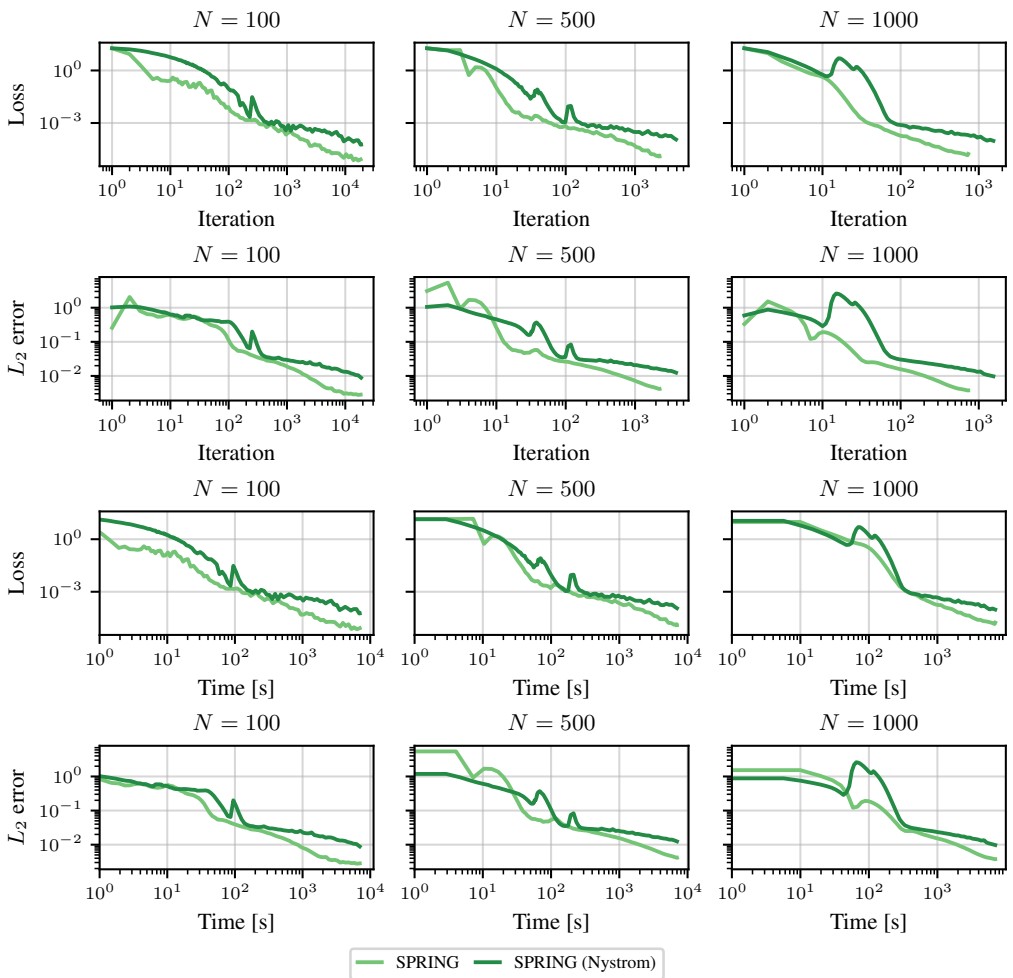

Figure 15: Training loss and evaluation $L^2$ error for learning the solution to a 100d Poisson equation over time and steps with large batch sized and fixed learning rate.

- **ENGD-W:** damping: $1.933\,629 \times 10^{-7}$
- **SPRING:** damping: $1.081\,840 \times 10^{-7}$, momentum: $8.500\,992 \times 10^{-1}$

**Search space details** The runs shown in Figure 16 were determined to be the best via a random search on the following search spaces which each optimizer given approximately the same total computational time ($\mathcal{U}$ denotes a uniform, and $\mathcal{LU}$ a log-uniform distribution):

- **ENGD-W:** damping: $\mathcal{LU}([1 \times 10^{-10}; 1 \times 10^{-1}])$
- **SPRING:** damping: $\mathcal{LU}([1 \times 10^{-10}; 1 \times 10^{-3}])$; momentum: $\mathcal{LU}([0.6; 0.999])$

### A.5.1 Fixed learning rate

We repeat the previous experiments, now adding a search space of $\mathcal{LU}([1 \times 10^{-1}; 1 \times 10^{-4}])$ for the learning rate, see Figure 17. We find that the best parameters are:

- **ENGD-W:** damping: $1.139\,970 \times 10^{-7}$, learning rate: $9.939\,225 \times 10^{-2}$
- **SPRING:** damping: $2.081\,775 \times 10^{-7}$, momentum: $9.078\,456 \times 10^{-1}$, learning rate: $8.663\,887 \times 10^{-2}$

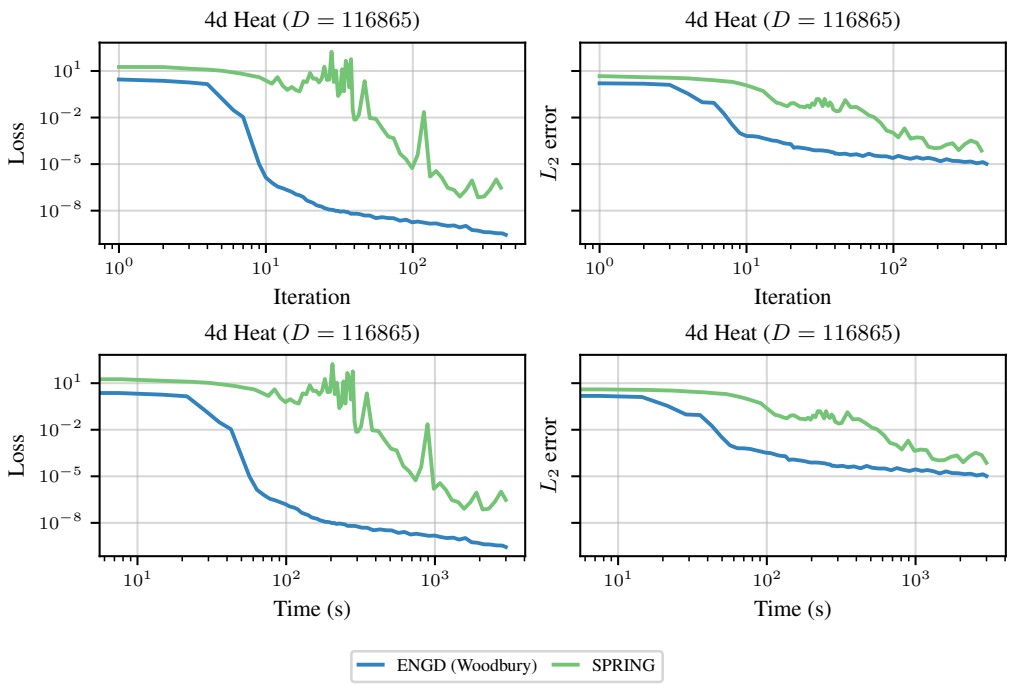

Figure 16: Training loss and evaluation $L^2$ error for learning the solution to the $4+1$d Heat equations over time and steps using random search.

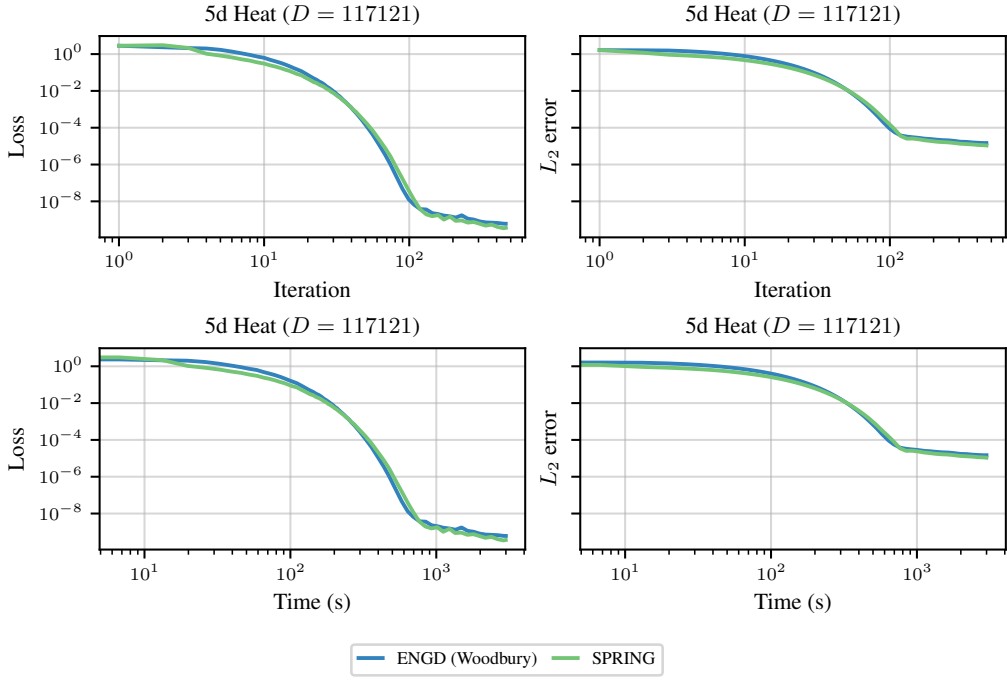

Figure 17: Training loss and evaluation $L^2$ error for learning the solution to a $4+1$d Heat equation over time and steps with fixed learning rate.

## A.6    9+1d Fokker-Planck equation in logarithmic space

**Setup**    For a given drift $\mu : [0,1] \times \mathbb{R}^d \to \mathbb{R}^d$ and diffusion coefficient $\sigma : [0,1] \to \mathbb{R}^{d \times d}$, the Fokker-Planck equation with initial probability density $p_0$ is given by

$$\partial_t p + \langle \nabla, \mu p \rangle - \frac{1}{2} \text{Tr}(\sigma \sigma^\top \nabla^2 p) = 0, \quad p(0) = p_0,$$

which is posed on $[0,1] \times \mathbb{R}^d$. Note that $p(\cdot, t)$ is a probaility density on $\mathbb{R}^d$ for all $t \in [0,1]$. We transform the above equation into logarithmic space via $q = \log p$. Then, $q$ solves

$$\partial_t q + \langle \nabla, \mu \rangle + \langle \nabla q, \mu \rangle - \frac{1}{2} ||\sigma^\top \nabla q||^2 - \frac{1}{2} \text{Tr}(\sigma \sigma^\top \nabla^2 q) = 0, \quad q(0) = \log p_0,$$

For our experiment, we set $\mu = (t, x) = -\frac{1}{2}x$ and $\sigma = \sqrt{2}I \in \mathbb{R}^{d \times d}$ and replace the unbounded domain by $[0,1] \times [-5, 5]^d$. Finally, our solution $q_\star = \log p_\star$ where $p_\star(t, x) \sim \mathcal{N}(0, \exp(-t)I + (1 - \exp(-t))2I$. Our loss includes the PDE residual and its initial conditions. We use a tanh-activated five-layer MLP: $10 \to 256 \to 256 \to 128 \to 128 \to 1$ and use batch sizes of $N_\Omega = 3000$ and $N_{\partial\Omega} = 1000$. Each run has an allocation time budget of 6000 s.

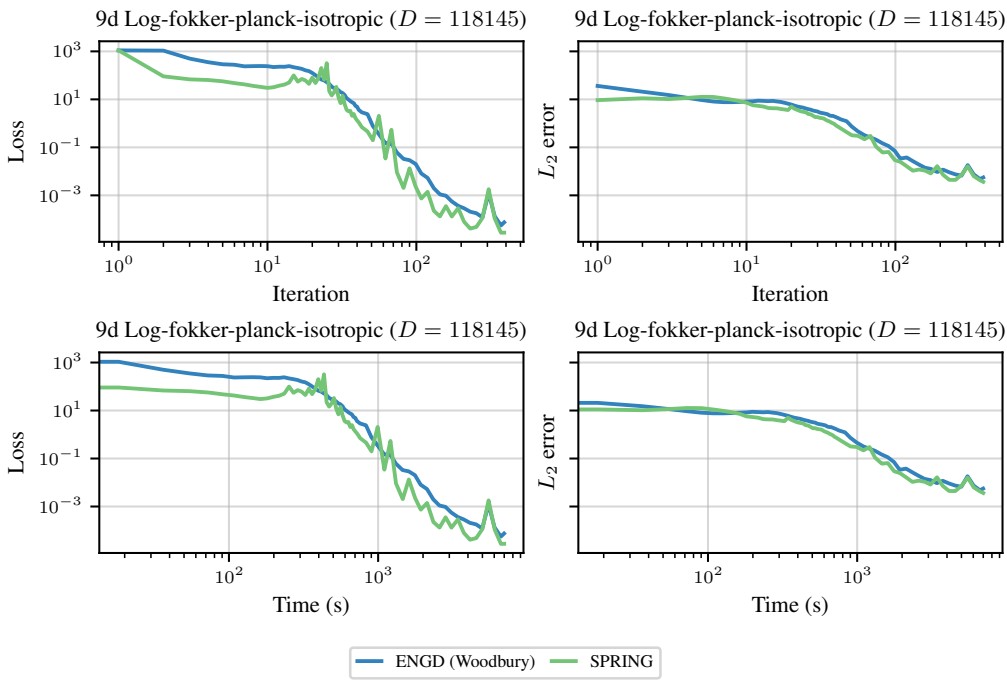

Figure 18: Training loss and evaluation $L^2$ error for learning the solution to the $9 + 1$d logarithmic Fokker-Planck equation over time and steps using random search.

**Best run details**    The runs shown in Figure 18 correspond to the following hyper-parameters:

- **ENGD-W:** damping: $1.558\,395 \times 10^{-10}$
- **SPRING:** damping: $7.511\,981 \times 10^{-2}$ momentum: $9.356\,251 \times 10^{-1}$

**Search space details**    The runs shown in Figure 18 were determined to be the best via a random search on the following search spaces which each optimizer given approximately the same total computational time ($\mathcal{U}$ denotes a uniform, and $\mathcal{LU}$ a log-uniform distribution):

- **ENGD-W:** damping: $\mathcal{LU}([1 \times 10^{-10}; 1 \times 10^{-1}])$
- **SPRING:** damping: $\mathcal{LU}([1 \times 10^{-10}; 1 \times 10^{-3}])$; momentum: $\mathcal{LU}([0.6; 0.999])$

### A.6.1 Fixed learning rate

We repeat the previous experiments, now adding a search space of $\mathcal{LU}([1 \times 10^{-1}; 1 \times 10^{-4}])$ for the learning rate, see Figure 19. We find that the best parameters are:

- **ENGD-W:** damping: $8.638\,985 \times 10^{-4}$; learning rate: $6.029\,401 \times 10^{-2}$
- **SPRING:** damping: $8.377\,655 \times 10^{-3}$; momentum: $9.760\,086 \times 10^{-1}$; learning rate: $4.473\,188 \times 10^{-2}$

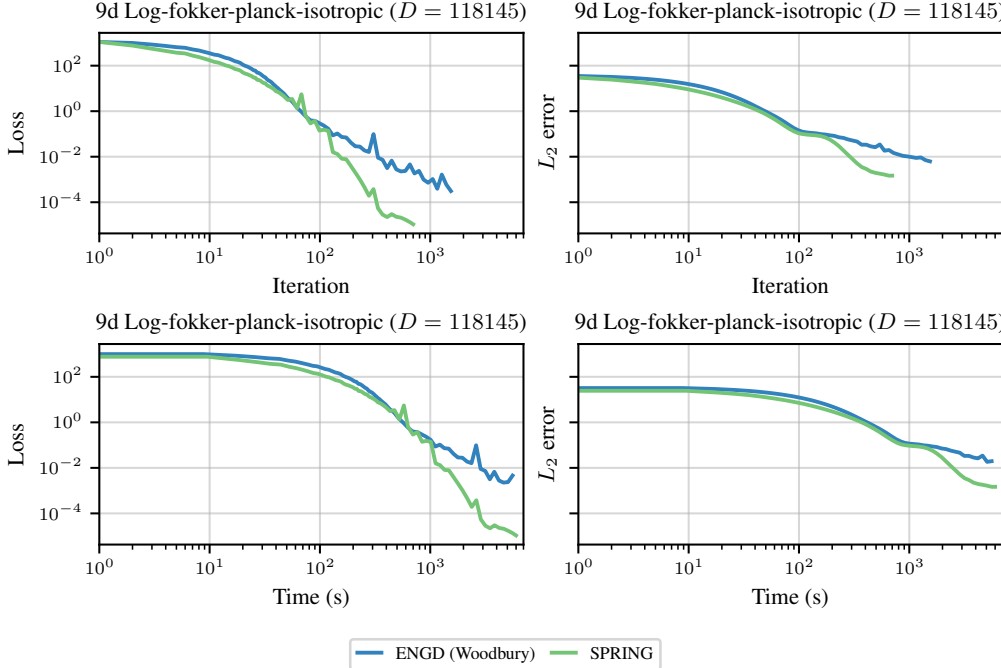

Figure 19: Training loss and evaluation $L^2$ error for learning the solution to the $9 + 1$d logarithmic Fokker-Planck equation over time and steps with fixed learning rate.

## B  GPU-efficient Implementation Benchmarking

We compare the performance of the traditional Nyström approximation [10], and our GPU-efficient proposal. Tests were performed on NVIDIA RTX 6000 GPUs (24 GiB RAM) with PyTorch's built-in timing routines, and the per-iteration execution time was averaged over 100 runs with 10 runs as warm-up. We show here a scaling analysis for $N = 5000$ and constant regularizer $\mu = 1.0 \times 10^{-7}$,

| Sketch size [% of $N$] | Avg. Time [$s$] | Peak Memory [$MiB$] | Speedup [$times$] |
|---|---|---|---|
| 20% | 0.1321 vs. **0.0099** | 233.9 vs. **165.2** | 13.29 |
| 40% | 0.6246 vs. **0.0253** | 395.4 vs. **234.6** | 24.70 |
| 60% | 2.1137 vs. **0.0461** | 599.6 vs. **312.5** | 45.90 |
| 80% | 4.7717 vs. **0.0725** | 856.8 vs. **395.0** | 65.41 |

Table 1: **Nyström approximation performance.** *Comparison between the standard Nyström method [10] and our **GPU-efficient Nyström** (boldfaced values). We report average runtime (s), peak memory (MiB), and speedup (×, defined as standard/GPU-efficient) across sketch sizes (% of $N$). Our method delivers substantial acceleration (≈13–65×) and consistently lower peak memory, with both speedups and memory savings increasing with sketch size.*

