# OpenReview forum: "Improving Energy Natural Gradient Descent through Woodbury, Momentum, and Randomization"
_NeurIPS.cc/2025/Conference — NeurIPS 2025 poster_

### Official Review · Reviewer_11hA · 2025-06-13

**Clarity:** 3
**Significance:** 3
**Originality:** 2
**Rating:** 4
**Confidence:** 3

**Summary:**

The paper applies a natural gradient optimizer (SPRING) which was originally developed for Variational Monte Carlo (VMC) to PINNs.
They find that similar to VMC, natural gradient descent converges substantially faster, and that similar to VMC using the Woodbury matrix identity substantially speeds up the optimizer.
The authors additionally try to accelerate the optimizer further by using a randomized Nystrom approximation (with a potentially more GPU-friendly implementation), but find that the randomized approximation is not accurate enough for practical use.
The optimizers are tested on the toy problem of the Poisson equation for 5-100D.

**Questions:**

- What is the performance of KFAC for these systems? In particular how does it fare in the regime of large number of parameters P and large batch size N, where presumable both ENGD (scaling poorly with P) and SPRING (scaling poorly with N) are expensive?
- Is there a setting where randomized variants of SPRING are superior to deterministic variants, potentially on a more challenging problem which may require larger batch-sizes?
- Does the faster convergence and runtime of SPRING unlock new applications of PINNs? Can the authors present results on a non-trivial system (e.g. fluid dynamics, inverse problems) where SPRING allows to improve the SOTA?


Minor questions:

- What exactly are the "Naive" optimizers in Fig. 4,5? Is it SVD-based randomized Nystrom? If yes, why does it converge at basically the same rate as your Cholesky-based Nystrom if the latter one is "an order of magnitude faster on GPU"?
- The authors write on line 189 "Such a sketch-and-solve approach has significant pitfalls [...]". What are these pitfalls? Is it "just" an inaccurate approximation if the target rank is less than the effective rank?

**Ethical Concerns:**

["NO or VERY MINOR ethics concerns only"]

**Final Justification:**

I had 2 primary concerns with the paper:
a) Limited evaluation of the method on simple systems
b) Limited novelty

The authors have addressed a) by providing data on 2 more systems and also compare against KFAC.
Point b) is still a concern for me because Natural Gradient is already widely used in PINNs and simply transferring the Woodbury acceleration trick from VMC seems like a modest contribution.
In summary I have increased my rating from 3 to 4 to reflect the authors' improvements.

**Limitations:**

The limitations section currently only addresses the limitations of the randomized approach.
I encourage the authors to also address the limitation of having evaluated the optimizers only on very small/simple test systems.

**Quality:**

2

**Strengths And Weaknesses:**

Strengths
-------------

- The paper transfers an established "trick" to accelerate natural gradient from one field (VMC) to an adjacent field (PINNs). This may be of value to PINNs practitioners, who are unfamiliar with the optimizer literature.
- On the investigated problems (5D and 100D Poisson equation on a hypercube), SPRING consistently outperforms other optimizers thanks to the convergence rate of natural gradient descent with the speed obtained through the Woodbury formulation
- The paper is well written and easy to read


Weaknesses
-----------------

- The paper presents minimal novel contributions, and only applies an existing optimizer to PINNs. The novel contribution (randomized Nystrom) is shown to be inferior to the deterministic variant in terms of loss obtained with a given compute budget.
- The method is only evaluated on extremely simple toy systems (linear PDE with explicit solution)
- The paper claims that SPRING surpasses KFAC performance, but does not contain any data on KFAC


Minor details
----

- The 2 shades of blue in Fig. 4,9,10 are very hard to distinguish, please consider slightly more distinct colors

---

> ### Author Rebuttal · Authors · 2025-07-31
>
> Dear Reviewer 11hA,
>
> Thanks a lot for your constructive feedback. Please find our answers below and let us know if you have follow-up questions.
>
> ---
>
> > 1. The paper presents minimal novel contributions, and only applies an existing optimizer to PINNs. The novel contribution (randomized Nystrom) is shown to be inferior to the deterministic variant in terms of loss obtained with a given compute budget.
>
> We respectfully disagree with both A) the implication that randomized Nyström is the only novel contribution of our work and B) the implication that the contribution regarding the randomized Nyström method is not significant due to its underperformance.
>
> Regarding point A), applying the Woodbury matrix identity and the SPRING algorithm within the context of ENGD for PINNS are both novel contributions that yield very encouraging results and have the potential to make an impact on the field of PINNs. Although these contributions do not come in the form of entirely new algorithms, they align with the statement in the NeurIPS reviewer guidelines that **"originality does not necessarily require introducing an entirely new method. Rather, a work that provides novel insights by evaluating existing methods, or demonstrates improved efficiency, fairness, etc. is also equally valuable."**
>
> Regarding point B), the fact that the randomized Nyström approximation does not improve the performance of ENGD is quite surprising and only holds as a result of the algorithmic improvements that we have introduced to the baseline ENGD implementation. In fact, one recent work applied a sketching method to reduce the cost of the *original* ENGD algorithm and found that relative to that inferior baseline, sketching is highly beneficial [2]. Our work thus overturns previous results by showing that, when ENGD is implemented efficiently using the Woodbury matrix identity, randomization is no longer benefcial.
>
> > 2. Is there a setting where randomized variants of SPRING are superior to deterministic variants, potentially on a more challenging problem which may require larger batch-sizes?
>
> We attempted but failed to identify any scenarios where randomized methods are superior.  As discussed above, this is a surprising finding of our paper which is tightly connected to the algorithmic improvements that we introduced to the deterministic algorithm.
>
> ---
>
> > 3. The method is only evaluated on extremely simple toy systems (linear PDE with explicit solution)
>
>
> Echoing our response to Reviewer rnPC, this is a fair point. To address your concern, **we evaluated two more PDEs** (to be included in the next revision): We consider the 4+1d Heat equation and the 9+1d logarithmic Fokker-Planck equation (a challenging nonlinear equation). For both equations, we use the same batch size and computational budget as in [1] sampling one batch at each step, their neural net with 100k parameters, and tune all algorithms with random search.
>
> The results are as follows:
>
> | Optimizer | $L_2$-error (Heat, 3000s) | $L_2$-error (log-FP, 6000s) |
> | --------- | ------------------ | --------------------- |
> | ENGD-W    | **1.0e-5**        | 1.1e-2                |
> | SPRING    | 1.1e-5             | **1.5e-3**            |
> | KFAC (taken from [1]) | 2.1e-5 | 2.6e-2                |
> | Adam (taken from [1]) | 1.6e-3 | 1.8e-1                |
>
> From these additional experiments we conclude:
> 1. Both ENGD-W and SPRING achieve state-of-the-art accuracies as in our previous experiments, and the advantage of SPRING is confirmed on the more challenging log-FP equation.
> 2. In comparison to KFAC [1], we first note that ENGD-W and SPRING now scale to this setting (before, ENGD in parameter space would run out of memory due to the neural net's size). Furthermore, we find that **both ENGD-W and SPRING reach 2 to 15-times smaller $L_2$-error than KFAC**.
>
> These findings demonstrate that the applicability of our methods is not restricted to toy examples of linear PDEs with explicit solutions. Let us know if this makes sense to you and if you have follow-up questions.
>
> ---
>
> > 4. The paper claims that SPRING surpasses KFAC performance, but does not contain any data on KFAC. What is the performance of KFAC for these systems?
>
> See the previous response for two new examples where both ENGD and SPRING outperform KFAC. Additionally, for the 100D Poisson equation, KFAC has previously achieved an $L_2$-error of 4.3e-3 with a time budget of 10,000 seconds as shown in [1], while in our work SPRING achieves an $L_2$-error of 3.6e-3 with a smaller budget of only 7,000 seconds and identical hardware.
>
> ---
> > 5. In particular how does [KFAC] fare in the regime of large number of parameters P and large batch size N, where presumable both ENGD (scaling poorly with P) and SPRING (scaling poorly with N) are expensive?
>
> You are right that KFAC scales more favourably in batch size than ENGD-W and SPRING. Our comparison focuses on batch sizes that are used in practise (all settings are from existing papers), and we do not observe this advantage. However, we will make this contrast to KFAC more explicit in the manuscript; thanks for pointing it out.
>
> ---
>
> > 6. Does the faster convergence and runtime of SPRING unlock new applications of PINNs? Can the authors present results on a non-trivial system (e.g. fluid dynamics, inverse problems) where SPRING allows to improve the SOTA?
>
> We view this as an important question for future works; the current work is focused on developing and demonstrating methods rather than tackling specific applications. Nonetheless, we refer the reviewer to the new heat and log-Fokker-Planck equations, which present new, non-trivial settings.
>
> ---
>
> > 7. What exactly are the "Naive" optimizers in Fig. 4,5? Is it SVD-based randomized Nystrom? If yes, why does it converge at basically the same rate as your Cholesky-based Nystrom if the latter one is "an order of magnitude faster on GPU"?
>
> This is a very good question and we apologize for the confusion. The figures were intended to show the SVD-based method but actually in their current form the naive methods refer to a fully naive Nyström method which approximates $A$ directly as $(A\Omega)(\Omega^T A \Omega)^{-1}(A\Omega)^T$. This naive implementation is known within the linear algebra community to be unstable, as discussed in [3] and others. In our own (intended to be internal) experiments this method worked surprisingly well (similar to the GPU-efficient stable algorithm), but we do not feel comfortable recommending it due to the known instability, which could become a problem in other settings. Unfortunately, the fix proposed by Frangella et al is not GPU-efficient and performs much worse than both of the methods currently pictured in Figures 4 and 5. This lead us to develop our own Nyström approximation that is both stable **and** GPU-efficient.
>
> Regarding the figures, it was a mistake to include this naive method and we will remove it in the final version. We will instead show the standard (SVD-based) stable algorithm of [3], so that the relative advantage of our proposed GPU-efficient Nyström method becomes clear. We again apologize for the confusion.
>
> ---
>
> > 8. The authors write on line 189 "Such a sketch-and-solve approach has significant pitfalls [...]". What are these pitfalls? Is it "just" an inaccurate approximation if the target rank is less than the effective rank?
>
> Yes, the specific issue is that with sketch-and-solve, it is impossible to obtain a highly accurate solution without taking the sketch size to be extremely large, which eliminates any computational savings. We will clarify this point in the manuscript.
>
>
> # References
>
> [1] Dangel, F., Müller, J., and Zeinhofer, M. Kronecker-factored approximate curvature for physics-informed neural networks. Conference on Neural Information Processing Systems, 2024.
>
> [2] Mckay, M. B., Kaur, A., Greif, C., and Wetton, B. Near-optimal sketchy natural gradients for physics-informed neural networks. In Forty-second International Conference on Machine Learning, 2025
>
> [3] Frangella, Z., Tropp, J. A., and Udell, M. Randomized Nyström preconditioning. SIAM Journal on Matrix Analysis and Applications, 44(2):718–752, 2023.

---

> > ### Comment · Reviewer_11hA · 2025-08-05
> >
> > I thank the authors for their clarifications and in particular appreciate the additional numerical experiments as well as the comparison against KFAC.
> > I suggest to include the discussion regarding sketching / McKay into the manuscript, showing that sketching only accelerates the non-Woodbury baseline.
> >
> > I will slightly raise my score.

---

### Official Review · Reviewer_nNgV · 2025-07-02

**Clarity:** 3
**Significance:** 2
**Originality:** 2
**Rating:** 5
**Confidence:** 3

**Summary:**

This paper introduces techniques for doing second-order optimization for PINN. The method connects previous related methods and theoretical results like ENGD, SPRING, Woodbury's identity, Nystrom approximation, and has demonstrated good performance in terms of convergence speed and accuracy for solving PINNs. Specifically, the paper (1) adapted ENGD and SPRING for the setting of PINN; (2) introduced Woodbury's identity in ENGD to reduce computational and memory cost; (3) proposed a GPU-efficient randomized Nystrom approximation, which is used to randomize ENGD and SPRING. The paper also analyzes the effective dimension that characterizes the limit of randomization. Overall, the paper presents a set of improvements to second-order optimization in PINNs.

**Questions:**

- Does your method works well with larger networks? All the experiments were conducted the MLP only.
- For the GPU-efficient randomized Nystrom, do you have any concrete comparsion against the original versino? For example, you could compare wall-clock time and memory usage profiles for different problem sizes, scaling behavior as problem dimensions increase, and other GPU utilization statistics.

**Ethical Concerns:**

["NO or VERY MINOR ethics concerns only"]

**Final Justification:**

I raised my score due to the following from the rebuttal
- The additional experiment indeed demonstrates the efficiency of the GPU-efficient Nystrom approximation
- The author confirms that the method works beyond MLP

**Quality:**

3

**Strengths And Weaknesses:**

Strength:
- The method is principled. All the techniques used were mathematically grounded.
- Main results were good: Woodbury version of ENGD has roughly the same convergence curve but ran faster, and the negative result of randomization is also interesting
- Experiments were thorough.
- Technical writing was on point. The presentation of each technical detail was described with sufficient detail without overcrowding the paper.

Weakness:
- The originality of the paper is a bit lacking. ENGD, SPRING, Woodbury's identity, and Nystrom approximation are all established techniques. The main innovation point of the paper seems to be applying these techniques to the specific setting of PINNs.
- There are a lot of concepts being shown here. I feel like the organization of the paper could be improved. The abstract could be a bit more descriptive so that the reader has a clearer expectation. The contribution could be clearly stated there.

---

> ### Author Rebuttal · Authors · 2025-07-31
>
> Dear Reviewer nNgV,
>
> Thank you for your constructive feedback. Please find our answers below and let us know if you have follow-up questions.
>
> ---
>
> > 1. The originality of the paper is a bit lacking. ENGD, SPRING, Woodbury's identity, and Nystrom approximation are all established techniques. The main innovation point of the paper seems to be applying these techniques to the specific setting of PINNs.
>
> We respectfully disagree with the implied concern regarding a lack of originality. According to the NeurIPS reviewer guidelines, originality is not limited to introducing entirely new methods. Rather, *“a work that provides novel insights by evaluating existing methods, or demonstrates improved efficiency, fairness, etc. is also equally valuable.”* Applying Woodbury's identity, SPRING, and the Nystrom approximation to ENGD for the first time aligns with this broader definition.
>
> ---
>
> > 2. For the GPU-efficient randomized Nystrom, do you have any concrete comparsion against the original version? For example, you could compare wall-clock time and memory usage profiles for different problem sizes, scaling behavior as problem dimensions increase, and other GPU utilization statistics.
>
> To corroborate our results, we generate random matrices of size $N=5000$ and compare the standard Nyström implementation of [1] with our GPU-efficient Nyström approximation on a GPU. We run the benchmarking procedure and present the results in the following table (we use the pattern *Standard Nystrom vs. GPU-efficient Nystrom*):
>
> | Sketch size [% of N]    | Avg. Time: Standard vs. GPU-efficient Nyström[s] | Peak Memory [MiB] | Speedup [times] |
> | ---- | ------------------ | --------------- | ----- |
> | 20% | 0.1321 vs. 0.0099  | 233.9 vs. 165.2 | **13.29** |
> | 40% | 0.6246 vs. 0.0253  | 395.4 vs. 234.6 | **24.70** |
> | 60% | 2.1137 vs. 0.0461  | 599.6 vs. 312.5 | **45.90** |
> | 80% | 4.7717 vs. 0.0725  | 856.8 vs. 395.0 | **65.41** |
>
> As you can see from the results, **the GPU-efficient implementation is at least an order of magnitude faster**. We will include this evidence, and more ablations, in our manuscript.
>
> ---
>
> > 3. Does your method works well with larger networks? All the experiments were conducted the MLP only.
>
> Yes! Since our methods are linear in $P$ and architecture-agnostic (unlike KFAC), there is no barrier to extending them to networks of larger sizes or different types. The only reason we use MLPs is to keep the networks the same as in other papers on ENGD to enable fair comparisons with previous works.
>
> ---
>
> > 4. I feel like the organization of the paper could be improved. The abstract could be a bit more descriptive so that the reader has a clearer expectation. The contribution could be clearly stated there.
>
> We appreciate the feedback and we would like to make the paper as clear as possible. To help us do so, could you elaborate on what is unclear to you in the current write-up, or what changes you would suggest in order to improve the organization of the paper?
>
> # References
>
> [1] Frangella, Z., Tropp, J. A., and Udell, M. Randomized Nyström preconditioning. SIAM Journal on Matrix Analysis and Applications, 44(2):718–752, 2023.

---

> > ### Comment · Reviewer_nNgV · 2025-08-05
> >
> > I thank the authors for responding to my comments thoroughly and running the additional experiments. I will raise my score slightly.

---

### Official Review · Reviewer_ZTVE · 2025-07-02

**Clarity:** 3
**Significance:** 4
**Originality:** 3
**Rating:** 5
**Confidence:** 3

**Summary:**

The paper presents a number of improvements for training physics-informed neural networks using natural gradient methods.  Precisely:

1. using the Morisson-Woodbury formula to reduce the computational complexity of the matrix inversion from $P^3$ (where $P$ is the number of parameters) to $PN^2$ (where $N$ is the number of vectors in the grid to enforce the DE and boundary conditions).

2. introducing a novel momentum term into borrowed from the SPRING algorithm.

3. further accelerating the matrix inversion using a novel, GPU-efficient Nystrom approximation method to avoid the costly (on a GPU) SVD calculation in the standard Nystrom approximation algorithm.

**Questions:**

See strengths and weaknesses section.

**Ethical Concerns:**

["NO or VERY MINOR ethics concerns only"]

**Final Justification:**

The author has addressed my concerns, and I am happy to keep my recommendation as accept.

**Limitations:**

Yes.

**Quality:**

3

**Strengths And Weaknesses:**

I thought that the paper was quite well written overall. It was relatively easy to follow, the contributions were (mostly) clearly described, and the claims made were mostly adequately explained/motivated and experimentally demonstrated. And the experimental results quite clearly demonstrated the benefits in the contribution.

On the negative side, I think that the second contribution could be more clearly explained. The term $\phi$ in equation (6) just "appears" out of the blue without explanation, leaving the reader to search further afield for clarification. Moreover while I get that (6) is just the usual regularized least-squares expression, what I'm missing is why this is to be preferred over any number of alternative regularization schemes that would, potentially, correspond to different momentum terms?

I am also a tad confused by (9). It appears from algorithm 2 that the $\lambda {\bf I}$ term is already incorporated into the GPU-efficient Nystrom calculation, yet for some reason the right-side of (9) reads $(nys (J_k J_k^T) + \lambda I)^{-1}$ and not $nys (J_k J_k^T + \lambda I )^{-1}$. Is this a typo, or have I misread the text?

The results section, however, quite clearly shows the benefit of the algorithm from a computation perspective, so in balance I believe the strengths outweigh the weaknesses here.

---

> ### Author Rebuttal · Authors · 2025-07-31
>
> Dear Reviewer ZTVE,
>
> Thank you for your constructive feedback. Please find our answers below and let us know if you have follow-up questions.
>
> ---
>
> > 1. The term $\phi$ in equation (6) just "appears" out of the blue without explanation, leaving the reader to search further afield for clarification. Moreover while I get that (6) is just the usual regularized least-squares expression, what I'm missing is why this is to be preferred over any number of alternative regularization schemes that would, potentially, correspond to different momentum terms?
>
> In equation (6), the variable $\phi$ is just a dummy variable that we optimize over, so it holds no direct meaning. Furthermore, the addition of $\lambda I$ on the left-hand side of (6) is standard within the natural gradient literature and is equivalent to adding an $L_2$ penalty on the right-hand side.
>
> Aside from this, maybe you are referring to the addition of the SPRING momentum term in equation (7)? The intution of equation (7) is to encourage the update direction to be close to the previous one, but it is true that this could be formalized in many ways. We simply use the same choice as the original SPRING paper [1] since this choice has been demonstrated to be effective.
>
> ---
>
> > 2. I am also a tad confused by (9). It appears from algorithm 2 that the  term is already incorporated into the GPU-efficient Nystrom calculation, yet for some reason the right-side of (9) reads $(\text{nys}(J_k J_k^T) + \lambda I)^{-1}$ and not $\text{nys}(J_k J_k^T + \lambda I)^{-1}$. Is this a typo, or have I misread the text?
>
> Thanks for catching this. There is a typo on Algorithm 2, which creates the confusion. In the "Ensure" statement of Algorithm 2, it should read:
> -  $\hat{A}_{\text{nys}} = B B^T$ on the left, and
> - $(\hat{A}_{\text{nys}} + \lambda I)^{-1}v = \dots$ on the right.
>
> We will rectify the error in the revised version.
>
> # References
> [1] Goldshlager, G., Abrahamsen, N., and Lin, L. A Kaczmarz-inspired approach to accelerate the optimization of neural network wavefunctions. Journal of Computational Physics, 516: 113351, 2024

---

> > ### Comment · Reviewer_ZTVE · 2025-08-05
> >
> > Thank you for your response - I am happy to keep my recommendation of accept.

---

### Official Review · Reviewer_rnPC · 2025-07-03

**Clarity:** 3
**Significance:** 2
**Originality:** 2
**Rating:** 4
**Confidence:** 3

**Summary:**

The authors use recently developed SPRING algorithm to address the optimization problems in Physics Informed Neural Networks (PINNs). The authors discuss the implementation of SPRING and propose a randomized algorithm to further accelerate the speed. The authors conduct several experiments to demonstrate the effectiveness of SPRING algorithm.

**Questions:**

If it is possible to implement other kinds of second-order optimizer with woodbury identity?

**Ethical Concerns:**

["NO or VERY MINOR ethics concerns only"]

**Final Justification:**

The reviewer acknowledges that the author's response has adequately addressed the second weakness previously identified. However, the scope of equations studied in the paper remains limited. Therefore, the reviewer recommends a borderline acceptance of this paper.

**Limitations:**

yes

**Paper Formatting Concerns:**

No concern

**Quality:**

3

**Strengths And Weaknesses:**

Strengths:

1. The authors make a valuable attempt to transfer insights and methodologies from the VMC community to the PINN community, which may boost both fields.

2. The introduction of a randomized variant of the SPRING algorithm offers the potential for improved accuracy-efficiency trade-offs, which may benefit practical applications.

Weaknesses:

1. The experimental evaluation is limited to the Poisson equation, which is relatively simple and may not sufficiently demonstrate the broader effectiveness or advantages of SPRING within the PINN framework. To convincingly establish its utility, the method should be tested on more challenging PDEs, such as the Navier–Stokes equations.

2. The paper directly applies the SPRING algorithm, originally designed for VMC problems, to PINN settings without adequately discussing the fundamental differences and similarities between these domains. For instance, VMC methods draw samples according to a distribution proportional to the square of the neural network, making stochastic reconfiguration (SR)-based methods natural and justified. In contrast, PINNs typically use samples drawn from distributions independent of the neural network, raising questions about why SR-based methods remain effective. A deeper theoretical or empirical discussion on this point is needed to clarify the applicability of SPRING to PINNs.

---

> ### Author Rebuttal · Authors · 2025-07-31
>
> Dear Reviewer rnPC,
>
> Thanks a lot for your constructive feedback. Please find our answers below and let us know if you have follow-up questions.
>
> ---
>
> > 1. The experimental evaluation is limited to the Poisson equation, which is relatively simple and may not sufficiently demonstrate the broader effectiveness or advantages of SPRING within the PINN framework. To convincingly establish its utility, the method should be tested on more challenging PDEs, such as the Navier–Stokes equations.
>
> This is a fair point. To address your concern, **we evaluated two more PDEs** (to be included in the next revision): We consider the 4+1d Heat equation and the 9+1d logarithmic Fokker-Planck equation (a challenging non-linear equation). For both equations, we use the same batch size and computational budget as in [1] and the same neural net with 100k parameters. With these settings we tune ENGD-W and SPRING with random search and compare the results to the results of Adam and KFAC from [1].
>
> The results are as follows:
>
> | Optimizer | $L_2$-error (Heat, 3000s) | $L_2$-error (log-FP, 6000s) |
> | --------- | ------------------ | --------------------- |
> | ENGD-W    | **1.0e-5**        | 1.1e-2                |
> | SPRING    | 1.1e-5             | **1.5e-3**            |
> | KFAC (taken from [1]) | 2.1e-5 | 2.6e-2                |
> | Adam (taken from [1]) | 1.6e-3 | 1.8e-1                |
>
> From these additional experiments we conclude:
> 1. Both ENGD-W and SPRING achieve state-of-the-art accuracies as in our previous experiments, and the advantage of SPRING is confirmed on the more challenging log-FP equation.
> 2. In comparison to KFAC [1], we first note that ENGD-W and SPRING now scale to this setting (before, ENGD in parameter space would run out of memory due to the neural net's size). Furthermore, we find that **both ENGD-W and SPRING reach 2 to 15-times smaller $L_2$-error than KFAC**.
>
> These new results demonstrate that the applicability of our methods is not restricted to the setting of the Poisson equation.
> Let us know if this makes sense to you and if you have follow-up questions.
>
> ---
>
> > 2. The paper directly applies the SPRING algorithm, originally designed for VMC problems, to PINN settings without adequately discussing the fundamental differences and similarities between these domains. For instance, VMC methods draw samples according to a distribution proportional to the square of the neural network, making stochastic reconfiguration (SR)-based methods natural and justified. In contrast, PINNs typically use samples drawn from distributions independent of the neural network, raising questions about why SR-based methods remain effective. A deeper theoretical or empirical discussion on this point is needed to clarify the applicability of SPRING to PINNs.
>
> Thank you for raising this point; it is indeed important to clarify why the techniques developed for the VMC community should be applicable to PINNs as well.
>
> **For both PINNs and VMC it is well-established that natural gradient methods are particularly well suited** [2, 3] and frequently outperform stochastic first-order methods like Adam. This motivates transfer between the fields.
>
> On a more technical level, **the probabilistic structure of the typical loss functions in PINNs and VMC are similar**: in both cases the loss function is defined by sampling from some distribution $p(R)$, which leads to the ENGD/SR update direction being the solution to a least-squares problem involving that same distribution $p(R)$, and as a result these algorithms and their variants such as SPRING should be implemented by sampling from $p(R)$ as well. In the case of VMC the distribution happens to be wavefunction dependent $p(R) = |\psi(R)|^2$, whereas in PINNs the distribution is a mixture of the uniform distributions over the interior and the boundary of the domain, but this difference does not affect the applicability of SPRING.
>
>
> ---
>
> >3. If it is possible to implement other kinds of second-order optimizer with woodbury identity?
>
> Yes, in fact Quasi-Newton methods like BFGS use the Woodbury identity by default, see for example the Wikipedia page of BFGS (here called Sherman-Morrison formula, a special case of the Woodbury formula). The vanilla Newton method (using the full Hessian) on the other hand cannot be accelarated with the Woodbury formula.
>
> # References
>
> [1] Dangel, F., Müller, J., and Zeinhofer, M. Kronecker-factored approximate curvature for physics-informed neural networks. Conference on Neural Information Processing Systems, 2024.
>
> [2] Müller, J., and Zeinhofer, M. Achieving High Accuracy with PINNs via Energy Natural Gradient Descent, International Conference on Machine Learning, 2023
>
> [3] Carleo and Troyer, Solving the Quantum Many-Body Problem with Artificial Neural Networks, Science, 2017

---

> > ### Comment · Reviewer_rnPC · 2025-08-05
> >
> > The reviewer thanks the authors for the feedback. The second point in weakness has been properly addressed by the response, so the reviewer would slightly increase the score.

---

### Decision · Program_Chairs · 2025-09-17

**Decision:**

Accept (poster)

**Comment:**

We thank the authors for their submission.  This work proposes a few practical ways to speed up second order optimization for physics-informed neural networks.

Reviewers agreed that the paper was well motivated, clearly written, and technically sound, commenting on the claims being well supported by empirical evidence generated.  It is clear that the resulting method has practical benefits.

Reviewers did voice the concern that this work lacked novelty, as it is a combination of existing methods.  Though this is a valid concern, the impressive performance results and careful empiricism of the work did generate some novel insights.  Additionally, though the approach may combine some well known methods, a degree of technical sophistication appears to be necessary to make them all work together.  Not to mention identifying PINNs as a suitable application to enjoy such benefits.  And the reviewers agreed — though most voiced this novelty concern, all reviewers see the value of the work and ultimately recommend acceptance.